# Missing-Data-Induced Phase Transitions in Spectral PLS for Multimodal Learning

## Abstract

Partial Least Squares (PLS) learns shared structure from paired data via the top singular vectors of the empirical cross-covariance (PLS-SVD), but multimodal datasets often have missing entries in both views. We study PLS-SVD under independent entry-wise missing-completely-at-random masking in a proportional high-dimensional spiked model. After appropriate normalization, the masked cross-covariance behaves like a spiked rectangular random matrix whose effective signal strength is attenuated by $\sqrt{\rho}$, where $\rho$ is the joint entry retention probability. As a result, PLS-SVD exhibits a sharp BBP-type phase transition: below a critical signal-to-noise threshold the leading singular vectors are asymptotically uninformative, while above it they achieve nontrivial alignment with the latent shared directions, with closed-form asymptotic overlap formulas. Simulations and semi-synthetic multimodal experiments corroborate the predicted phase diagram and recovery curves across aspect ratios, signal strengths, and missingness levels.

## 1. Introduction

Multimodal learning from paired data is a recurring theme in machine learning: Given two views $(X, Y)$ of the same entities, one aims to extract a shared low-dimensional structure that supports prediction, representation learning, or exploratory analysis. Canonical correlation analysis (CCA) (Hotelling, 1936) and Partial Least Squares (PLS) are classical linear approaches to this problem, differing primarily in whether they optimize correlation or covariance. PLS in particular remains widely used in high-dimensional applications due to its empirical stability and scalability (Rosipal & Krämer, 2006; Boulesteix & Strimmer, 2007). A common

[1]Anonymous Institution, Anonymous City, Anonymous Region, Anonymous Country. Correspondence to: Anonymous Author <anon.email@domain.com>.

Preliminary work. Under review by the International Conference on Machine Learning (ICML). Do not distribute.

spectral formulation of PLS computes the leading singular vectors of an empirical cross-covariance matrix (PLS-SVD), placing it within a broader family of singular-vector methods studied via random matrix theory (Benaych-Georges & Nadakuditi, 2012; Léger & Chatelain, 2025).

In practice, multi-view datasets are often not fully observed: Missing entries arise due to acquisition failures, measurement dropout, heterogeneous pipelines, or sensor outages. While missingness is ubiquitous, its impact on the recovery threshold and overlap behavior of two-view spectral estimators remains poorly understood. Even under the simplest missing-completely-at-random (MCAR) mechanism (Little & Rubin, 2002), the cross-covariance structure that drives PLS-SVD is corrupted in a nontrivial way when *both* views are subject to missing data ("masked"). This raises a simple question: *when does PLS-SVD still recover an informative shared component under masking in both $X$ and $Y$?*

We answer this question in a proportional high-dimensional spiked model with dual entry-wise MCAR masking. Let $\alpha_x = N/D_x$ and $\alpha_y = N/D_y$ denote aspect ratios, and let $m_x, m_y$ be the missing rates in the two views with joint retention probability $\rho = (1 - m_x)(1 - m_y)$. We study the leading left/right singular vectors $(\hat{u}, \hat{v})$ of a properly normalized cross-covariance and measure recovery through the squared overlaps $R_x^2 = (\hat{u}^\top u_0)^2$ and $R_y^2 = (\hat{v}^\top v_0)^2$ with the planted signal directions $(u_0, v_0)$. In the proportional limit, $R_x^2 \to r_x^2$ and $R_y^2 \to r_y^2$, where $(r_x^2, r_y^2)$ denote the asymptotic limits characterized in Theorem 3.2.

A key message is that dual missingness acts primarily as an attenuation of signal-to-noise rather than as a mere reduction in the effective number of samples. Concretely, masking induces an effective spike strength $\theta_{\text{eff}} = \sqrt{\rho}\,\theta$, shifting the recoverability boundary in the same spirit as the missing-data phenomenon identified for principal component analysis (PCA) (Ipsen & Hansen, 2019). This reduction turns PLS-SVD into a spiked rectangular singular-vector problem (Benaych-Georges & Nadakuditi, 2012) with a Baik–Ben Arous–Péché (BBP) type transition (Baik et al., 2005).

Our main result (Theorem 3.2) establishes that recovery is possible if and only if the signal strength exceeds a critical

threshold $\theta_{\text{crit}} = 1/((\alpha_x \alpha_y)^{1/4}\sqrt{\rho})$, with explicit closed-form formulas for the asymptotic overlaps above this threshold. When $\rho = 1$, these expressions reduce to the known spiked-rectangular overlap behavior (Benaych-Georges & Nadakuditi, 2012; Léger & Chatelain, 2025), while dual missingness increases the required signal strength by the factor $1/\sqrt{\rho}$.

Our main contributions are

- **Sharp dual-missingness phase transition for PLS-SVD:** we characterize when recovery is possible under entry-wise MCAR masking in both views, yielding the explicit threshold $\theta_{\text{crit}} = 1/((\alpha_x \alpha_y)^{1/4}\sqrt{\rho})$.

- **Closed-form recovery curves:** we derive explicit asymptotic formulas for the squared overlaps $(r_x^2, r_y^2)$ above the threshold as functions of $(\alpha_x, \alpha_y, \rho, \theta)$.

- **Replica-symmetric analysis via spiked-rectangular reduction:** we give a complete replica-symmetric (RS) derivation, showing missingness acts as an effective spike $\theta_{\text{eff}} = \sqrt{\rho}\,\theta$ and recovering the fully observed spiked-rectangular predictions when $\rho = 1$.

- **Empirical validation:** synthetic and semi-synthetic experiments (including TCGA BRCA and PBMC Multiome) corroborate the predicted phase boundary and overlap curves across aspect ratios, signal strengths, and missingness rates.

## 2. Related Work

**PLS, CCA, and high-dimensional two-view spectral methods.** CCA dates back to Hotelling (1936) and remains a canonical framework for extracting coupled linear structure from paired observations. PLS is a closely related alternative that maximizes cross-covariance and is widely used in modern high-dimensional applications (Rosipal & Krämer, 2006; Boulesteix & Strimmer, 2007). Both admit spectral formulations based on cross-covariance matrices, connecting two-view learning to singular-vector behavior in high dimensions (Benaych-Georges & Nadakuditi, 2012). Recent work has begun to analyze PLS-SVD in the proportional regime under fully observed spiked models (Léger & Chatelain, 2025). Our contribution addresses a different regime: we characterize when spectral PLS remains informative under *entry-wise missingness in both views*, yielding an explicit threshold and overlap curves as functions of $(\alpha_x, \alpha_y, \rho)$.

**Missing data in multivariate latent-variable models.** The statistical literature distinguishes MCAR/MAR/MNAR mechanisms and clarifies how missingness can distort estimation (Little & Rubin, 2002; Schafer & Graham,

2002). For single-view structure learning, probabilistic PCA (Roweis, 1998; Tipping & Bishop, 1999) provides a likelihood-based model that naturally accommodates incomplete observations via EM (Dempster et al., 1977). In the two-view setting, probabilistic formulations of PLS (el Bouhaddani et al., 2018) and variants developed for data integration (e.g., multi-omics) provide flexible models for inference and prediction (el Bouhaddani et al., 2022), while generalized CCA has also been adapted to missing values through algorithmic treatments (van de Velden & Takane, 2012). These methods focus on modeling and estimation procedures under missingness; in contrast, we target a sharp high-dimensional *recoverability boundary* for the classical spectral estimator PLS-SVD under dual MCAR masking.

**Phase transitions, spiked matrices, and replica methodology.** Sharp transitions are a hallmark of spiked random matrix models. The BBP phenomenon (Baik et al., 2005) and the singular-vector characterization of low-rank rectangular perturbations (Benaych-Georges & Nadakuditi, 2012) provide the natural mathematical baseline for PLS-SVD once the cross-covariance is cast in spiked form. Replica methods have long been used to analyze thresholds and overlaps for high-dimensional spectral inference (Biehl & Mietzner, 1993; Hoyle & Rattray, 2007) and connect to algorithmic perspectives such as message passing (Bayati & Montanari, 2011). More recently, Ipsen & Hansen (2019) used replica techniques to derive a phase transition for PCA with missing data and emphasized an effective signal-to-noise reduction viewpoint. Our work extends this line to two-view learning by giving a replica-symmetric derivation for PLS-SVD under dual MCAR masking, recovering fully observed spiked-rectangular formulas as a special case and isolating the missingness effect through the multiplicative attenuation $\theta_{\text{eff}} = \sqrt{\rho}\,\theta$.

## 3. Replica Analysis of PLS with Missing Data

### 3.1. Problem Setup

We study Partial Least Squares via Singular Value Decomposition (PLS-SVD) in a two-view model with missingness in both the design and the response. Let the *latent complete* (whitened) design be $X_\star \in \mathbb{R}^{N \times D_x}$ and assume

$$X_\star^\top X_\star = N I_{D_x}. \qquad (1)$$

Fix unit vectors $u_0 \in \mathbb{R}^{D_x}$ and $v_0 \in \mathbb{R}^{D_y}$, and generate the *latent complete* response

$$Y_\star = \theta(X_\star u_0)v_0^\top + Z, \quad Z_{ij} \overset{\text{i.i.d.}}{\sim} \mathcal{N}(0,1), \quad Z \perp X_\star, \quad (2)$$

with $\theta \geq 0$. We introduce independent missing-completely-at-random (MCAR) masks $S_x \in \{0,1\}^{N \times D_x}$ and $S_y \in \{0,1\}^{N \times D_y}$ with i.i.d. entries $(S_x)_{ij} \sim \text{Bernoulli}(1 - m_x)$ and $(S_y)_{ij} \sim \text{Bernoulli}(1 - m_y)$, independent of $(X_\star, Y_\star, Z)$. Define the retention probabilities $\rho_x := 1 -$

$m_x$, $\rho_y := 1 - m_y$, and $\rho := \rho_x \rho_y$. We observe missing-as-zero matrices

$$X := X_{\text{obs}} = S_x \odot X_\star, \qquad Y := Y_{\text{obs}} = S_y \odot Y_\star. \quad (3)$$

The PLS-SVD algorithm computes the top singular vectors of the empirical cross-covariance $\widehat{\Sigma}_{XY} := N^{-1} X^\top Y \in \mathbb{R}^{D_x \times D_y}$. Since singular vectors are invariant to nonzero scalar rescaling, we analyze the rescaled cross-covariance

$$C := \frac{1}{\sqrt{\rho}} \widehat{\Sigma}_{XY} = \frac{1}{N\sqrt{\rho}} X^\top Y. \quad (4)$$

PLS-SVD outputs

$$(\hat{u}, \hat{v}) \in \arg \max_{\|u\|=\|v\|=1} u^\top C v. \quad (5)$$

We measure recovery performance via squared overlaps

$$R_x^2 := (\hat{u}^\top u_0)^2, \qquad R_y^2 := (\hat{v}^\top v_0)^2. \quad (6)$$

We work in the proportional limit with fixed aspect ratios $\alpha_x := N/D_x$ and $\alpha_y := N/D_y$, both in $(0, \infty)$, as $N, D_x, D_y \to \infty$.

### 3.2. Reduction to the Spiked Model

Under dual MCAR masking, the observed cross-covariance reduces to a spiked rectangular Gaussian model with effective spike strength $\theta_{\text{eff}} = \sqrt{\rho}\,\theta$.

**Lemma 3.1** (Spiked Form under Dual Masking). *Under (1)–(3), define $C$ by (4). Then in the proportional limit,*

$$C = \theta_{\text{eff}}\, u_0 v_0^\top + \frac{1}{\sqrt{N}} W + o_{\mathbb{P}}(1/\sqrt{N}), \quad \theta_{\text{eff}} := \sqrt{\rho}\,\theta, \quad (7)$$

*where $W \in \mathbb{R}^{D_x \times D_y}$ has asymptotically i.i.d. $\mathcal{N}(0,1)$ entries.*

*Proof.* Expand (4) using (3) and (2):

$$C = \frac{1}{N\sqrt{\rho}} (S_x \odot X_\star)^\top \Big( S_y \odot \big(\theta(X_\star u_0) v_0^\top + Z\big) \Big).$$

The signal contribution has expectation

$$\mathbb{E}[C] = \frac{\theta}{N\sqrt{\rho}} \mathbb{E}\left[ (S_x \odot X_\star)^\top (S_y \odot (X_\star u_0) v_0^\top) \right]$$

$$= \sqrt{\rho}\,\theta\, u_0 v_0^\top,$$

where we used $\mathbb{E}[S_x \odot S_y] = \rho$ entrywise and $X_\star^\top X_\star = NI$. The centered remainder is a sum of independent terms with variance of order $1/N$. The dominant contribution comes from the masked Gaussian noise $S_y \odot Z$, which yields an asymptotically Gaussian matrix with i.i.d. $\mathcal{N}(0,1)$ entries after normalization by $\sqrt{\rho}$. Residual fluctuations from random masking of the signal are of smaller order and can be absorbed into $W$, establishing (7). $\qquad\square$

By Lemma 3.1, dual missingness reduces the effective signal-to-noise ratio by a factor of $\sqrt{\rho}$, which shifts the phase boundary accordingly.

### 3.3. Main Result: Phase Transition under Dual Masking

**Theorem 3.2** (PLS-SVD Phase Transition under Dual Masking). *Consider the spiked two-view model with whitened design $X_\star^\top X_\star = NI_{D_x}$, response $Y_\star = \theta(X_\star u_0) v_0^\top + Z$ with i.i.d. Gaussian noise, and independent MCAR masks with retention probabilities $\rho_x = 1 - m_x$, $\rho_y = 1 - m_y$. Let $\rho = \rho_x \rho_y$ denote the joint retention probability. As $N, D_x, D_y \to \infty$ with fixed aspect ratios $\alpha_x = N/D_x$ and $\alpha_y = N/D_y$, the squared overlaps of the leading PLS-SVD singular vectors with the planted directions converge in probability:*

$$r_x^2 = \begin{cases} 0, & \alpha_x \alpha_y \rho^2 \theta^4 \leq 1, \\ \dfrac{\alpha_x \alpha_y \rho^2 \theta^4 - 1}{\alpha_y \rho \theta^2 (\alpha_x \rho \theta^2 + 1)}, & \alpha_x \alpha_y \rho^2 \theta^4 > 1, \end{cases} \quad (8)$$

$$r_y^2 = \begin{cases} 0, & \alpha_x \alpha_y \rho^2 \theta^4 \leq 1, \\ \dfrac{\alpha_x \alpha_y \rho^2 \theta^4 - 1}{\alpha_x \rho \theta^2 (\alpha_y \rho \theta^2 + 1)}, & \alpha_x \alpha_y \rho^2 \theta^4 > 1. \end{cases} \quad (9)$$

*The critical threshold is*

$$\theta_{\text{crit}} = \frac{1}{(\alpha_x \alpha_y)^{1/4} \sqrt{\rho}}. \quad (10)$$

Recovery is possible if and only if $\theta > \theta_{\text{crit}}$. The remainder of this section analyze this result via replica analysis.

### 3.4. Replica Analysis

We provide a replica-symmetric derivation of the phase transition and overlap formulas. By Lemma 3.1, it suffices to analyze the spiked rectangular model

$$C = \theta_{\text{eff}}\, u_0 v_0^\top + \frac{1}{\sqrt{N}} W, \qquad W_{ij} \overset{\text{iid}}{\sim} \mathcal{N}(0,1),$$

since the top singular vectors of $C$ coincide with those of the observed cross-covariance. The derivation proceeds in four steps: (i) Gibbs formulation of the SVD objective, (ii) replica trick for the quenched free energy, (iii) reduction to order parameters via a replica-symmetric saddle point, and (iv) zero-temperature limit yielding closed-form overlaps.

**Gibbs Formulation and Source Fields.** We reformulate the SVD optimization problem as sampling from a Gibbs distribution at low temperature. Introduce the *inverse temperature* $\beta > 0$ and *source fields* $h_x, h_y \geq 0$. Define the Gibbs partition function

$$Z_\beta(h_x, h_y) := \int \delta(\|u\|^2 - 1)\, \delta(\|v\|^2 - 1)$$
$$\exp(\beta N\, u^\top C v) \exp(h_x\, u^\top u_0) \quad (11)$$
$$\times \exp(h_y\, v^\top v_0)\, du\, dv.$$

The integration is over all $u \in \mathbb{R}^{D_x}$ and $v \in \mathbb{R}^{D_y}$, with the delta functions restricting to the unit spheres. The term $\beta N\, u^\top C v$ is the scaled objective: $\beta$ is the inverse temperature, and the factor $N$ ensures extensive scaling. The

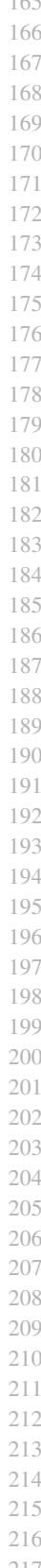

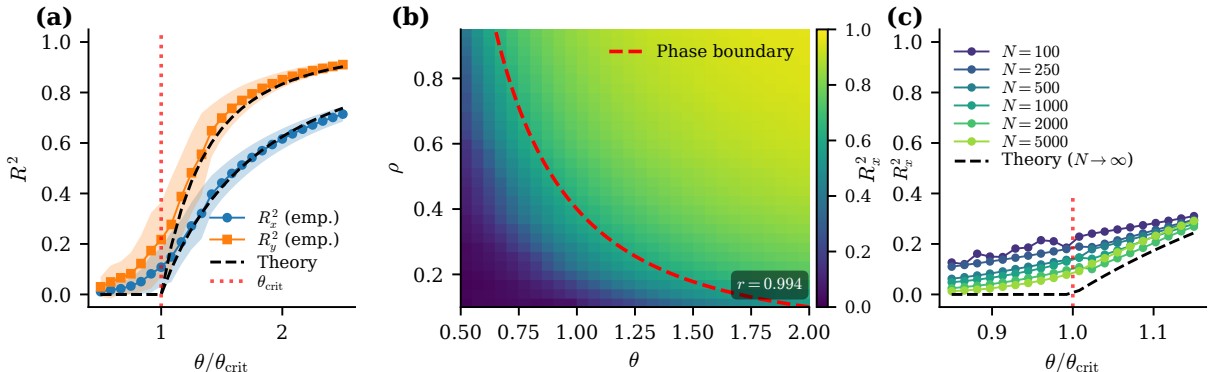

*Figure 1.* **Validation of the PLS-SVD phase transition theory under dual missingness. (a)** Phase transition in squared overlaps $R_x^2$ and $R_y^2$ as signal strength $\theta$ crosses the critical threshold $\theta_{\text{crit}} = 1/[(\alpha_x \alpha_y)^{1/4} \sqrt{\rho}]$ (red line). Empirical results (circles/squares with shaded error bands showing $\pm 1$ standard deviation) match theoretical predictions (dashed lines) from Theorem 3.2. **(b)** Phase diagram in $(\theta, \rho)$ space shows empirical overlap $R_x^2$ (heatmap) with theoretical phase boundary (red dashed curve) separating the subcritical regime (recovery impossible) from the supercritical regime (successful recovery). High correlation ($r = 0.994$) confirms theoretical accuracy. **(c)** Finite-size sharpening: as sample size $N$ increases from 100 to 5000, empirical overlaps converge to the sharp theoretical step function (black dashed line). Parameters: **(a)** $N = 1000$, $D_x = 200$, $D_y = 50$, $m_x = 0.3$, $m_y = 0.4$, 100 trials; **(b)** $N = 1000$, $D_x = 150$, $D_y = 120$, $30 \times 30$ grid, 30 trials; **(c)** $\alpha_x = \alpha_y = 2.5$, $m_x = m_y = 0.2$, 30 trials.

source fields $h_x, h_y$ break sign symmetry and allow extraction of overlaps via differentiation. As $\beta \to \infty$, the Gibbs measure concentrates on the SVD solution $(\hat{u}, \hat{v})$; the limit $h_x, h_y \downarrow 0^+$ breaks sign symmetry. The overlaps can be extracted via a stationary-point differentiation identity applied to the value function (see Appendix A.4), yielding $R_x^2 = (r_u^\star)^2$ and $R_y^2 = (r_v^\star)^2$ where $(r_u^\star, r_v^\star)$ are saddle-point "magnetizations".

**Replica Trick.** The typical free energy is obtained through

$$\mathbb{E}\left[\log Z_\beta(h_x, h_y)\right] = \lim_{w \to 0} \frac{1}{w} \log \mathbb{E}\left[Z_\beta(h_x, h_y)^w\right]. \quad (12)$$

For integer $w \geq 1$, the $w$-th power of the partition function is

$$Z_\beta(h_x, h_y)^w = \int \prod_{a=1}^{w} \delta(\|u^a\|^2 - 1)\, \delta(\|v^a\|^2 - 1)$$

$$\exp\left(\beta N \sum_{a=1}^{w} u^{a\top} C v^a\right) \exp\left(h_x \sum_{a=1}^{w} u^{a\top} u_0\right) \quad (13)$$

$$\times \exp\left(h_y \sum_{a=1}^{w} v^{a\top} v_0\right) dU\, dV,$$

where $u^a \in \mathbb{R}^{D_x}$ and $v^a \in \mathbb{R}^{D_y}$ for $a = 1, \ldots, w$ are the replica vectors, and $dU\, dV$ denotes integration over all replicas.

**Averaging over the Gaussian Noise.** Using Lemma 3.1, $C = \theta_{\text{eff}} u_0 v_0^\top + N^{-1/2} W$ with i.i.d. $W_{ij} \sim \mathcal{N}(0, 1)$.

Hence

$$\beta N \sum_{a=1}^{w} u^{a\top} C v^a = \beta N \theta_{\text{eff}} \sum_{a=1}^{w} (u^{a\top} u_0)(v^{a\top} v_0)$$

$$+ \beta \sqrt{N} \sum_{a=1}^{w} u^{a\top} W v^a. \quad (14)$$

Introduce the order parameters

$$(Q_u)_{ab} := u^{a\top} u^b, \qquad (Q_v)_{ab} := v^{a\top} v^b,$$
$$(r_u)_a := u^{a\top} u_0, \qquad (r_v)_a := v^{a\top} v_0. \quad (15)$$

Averaging over $W$ using standard Gaussian MGF identities (Appendix A.2) yields

$$\mathbb{E}\left[Z_\beta(h_x, h_y)^w\right] = \int \prod_{a=1}^{w} \delta(\|u^a\|^2 - 1)\, \delta(\|v^a\|^2 - 1)$$

$$\exp\left(\beta N \theta_{\text{eff}} \sum_{a=1}^{w} (r_u)_a (r_v)_a\right) \exp\left(\frac{\beta^2 N}{2} \operatorname{Tr}(Q_u Q_v^\top)\right) \quad (16)$$

$$\times \exp\left(h_x \sum_{a=1}^{w} (r_u)_a\right) \exp\left(h_y \sum_{a=1}^{w} (r_v)_a\right) dU\, dV,$$

where the order parameters $(Q_u)_{ab} = u^{a\top} u^b$, $(Q_v)_{ab} = v^{a\top} v^b$, $(r_u)_a = u^{a\top} u_0$, and $(r_v)_a = v^{a\top} v_0$ are implicit functions of the replica vectors.

**Saddle-Point Reduction.** The exponent in (16) depends on the replica vectors only through the order parameters $(Q_u, Q_v, r_u, r_v)$. Changing variables from replica vectors to these order parameters produces Jacobian factors $\det(Q_u - r_u r_u^\top)^{D_x/2}$ and $\det(Q_v - r_v r_v^\top)^{D_y/2}$ (Appendix A.3). Using $D_x = N/\alpha_x$ and $D_y = N/\alpha_y$, the

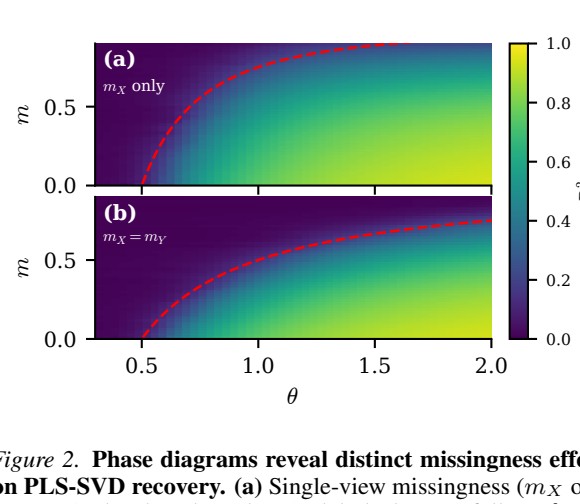

*Figure 2.* **Phase diagrams reveal distinct missingness effects on PLS-SVD recovery. (a)** Single-view missingness ($m_X$ only, $m_Y = 0$): the phase boundary (red dashed curve) follows $\theta_{\text{crit}} = 1/[(\alpha_x \alpha_y)^{1/4} \sqrt{1-m}]$, with retention probability $\rho = 1 - m$ degrading linearly. **(b)** Joint missingness ($m_X = m_Y = m$): the boundary follows $\theta_{\text{crit}} = 1/[(\alpha_x \alpha_y)^{1/4}(1-m)]$, reflecting quadratic retention degradation $\rho = (1-m)^2$. The steeper boundary demonstrates that joint missingness requires proportionally stronger signals for recovery. Heatmap shows empirical $R_x^2$ across $(\theta, m)$ parameter space. Parameters: $N = 800$, $D_x = D_y = 200$ ($\alpha_x = \alpha_y = 4$), $50 \times 50$ grid, 30 trials per point.

replicated action density is

$$
\begin{aligned}
S_w = {} & \frac{1}{2\alpha_x} \log \det(Q_u - r_u r_u^\top) \\
& + \frac{1}{2\alpha_y} \log \det(Q_v - r_v r_v^\top) \\
& + \beta\theta_{\text{eff}} \sum_{a=1}^{w} (r_u)_a (r_v)_a + \frac{\beta^2}{2} \operatorname{Tr}(Q_u Q_v^\top).
\end{aligned}
\tag{17}
$$

**Replica Symmetry and the $w \to 0$ Evaluation.** Under the replica-symmetric (RS) ansatz (Mézard et al., 1987; Nishimori, 2001):

$$
\begin{aligned}
(Q_u)_{aa} &= 1, \quad (Q_u)_{ab} = q_u \ (a \neq b), \\
(Q_v)_{aa} &= 1, \quad (Q_v)_{ab} = q_v \ (a \neq b), \\
(r_u)_a &= r_u, \quad (r_v)_a = r_v.
\end{aligned}
$$

Taking the $w \to 0$ limit (see Appendix A.5 for detailed derivation), the determinant term becomes

$$
\lim_{w \to 0} \frac{1}{w} \log \det(Q_u - r_u r_u^\top) = \log(1 - q_u) + \frac{q_u - r_u^2}{1 - q_u}, \tag{18}
$$

and analogously for $(q_v, r_v)$. The trace term simplifies to

$$
\lim_{w \to 0} \frac{1}{w} \operatorname{Tr}(Q_u Q_v^\top) = 1 - q_u q_v. \tag{19}
$$

**RS free energy.** Define the shorthand

$$
\phi(q, r) := \log(1 - q) + \frac{q - r^2}{1 - q}. \tag{20}
$$

Then the RS free-energy density becomes

$$
\begin{aligned}
\Phi_\beta(q_u, q_v, r_u, r_v; h_x, h_y) = {} & \frac{1}{2\alpha_x} \phi(q_u, r_u) + \frac{1}{2\alpha_y} \phi(q_v, r_v) \\
& + \frac{\beta^2}{2}(1 - q_u q_v) + \beta\theta_{\text{eff}} \, r_u r_v + h_x \, r_u + h_y \, r_v.
\end{aligned}
\tag{21}
$$

**Zero Temperature and Reduction to Two Overlaps.** As $\beta \to \infty$, $q_u, q_v \to 1$. Introduce susceptibilities $\chi_u := \beta(1 - q_u)$ and $\chi_v := \beta(1 - q_v)$ (see Appendix A.6 for detailed scaling analysis). Consider the rescaled objective $\Psi := \lim_{\beta \to \infty} \beta^{-1}\Phi_\beta$. This yields

$$
\begin{aligned}
& \Psi(r_u, r_v, \chi_u, \chi_v) \\
& = \frac{1 - r_u^2}{2\alpha_x \chi_u} + \frac{1 - r_v^2}{2\alpha_y \chi_v} + \frac{\chi_u + \chi_v}{2} + \theta_{\text{eff}} \, r_u r_v.
\end{aligned}
\tag{22}
$$

Optimizing over $\chi_u, \chi_v$ (see Appendix A.7) gives $\chi_u^\star = \sqrt{(1 - r_u^2)/\alpha_x}$ and $\chi_v^\star = \sqrt{(1 - r_v^2)/\alpha_y}$. Substituting back yields the reduced two-parameter objective

$$
\begin{aligned}
\Psi(r_u, r_v) = {} & \frac{\sqrt{1 - r_u^2}}{\sqrt{\alpha_x}} + \frac{\sqrt{1 - r_v^2}}{\sqrt{\alpha_y}} \\
& + \theta_{\text{eff}} \, r_u r_v, \qquad r_u, r_v \in [0, 1].
\end{aligned}
\tag{23}
$$

**Stationarity, Threshold, and Closed-Form Overlaps.** Stationarity implies

$$
\begin{aligned}
\theta_{\text{eff}} \, r_v &= \frac{1}{\sqrt{\alpha_x}} \frac{r_u}{\sqrt{1 - r_u^2}}, \\
\theta_{\text{eff}} \, r_u &= \frac{1}{\sqrt{\alpha_y}} \frac{r_v}{\sqrt{1 - r_v^2}}.
\end{aligned}
\tag{24}
$$

Squaring yields

$$
\frac{r_u^2}{1 - r_u^2} = \alpha_x \theta_{\text{eff}}^2 \, r_v^2, \quad \frac{r_v^2}{1 - r_v^2} = \alpha_y \theta_{\text{eff}}^2 \, r_u^2. \tag{25}
$$

Multiplying (for $r_u, r_v > 0$) gives

$$
(1 - r_u^2)(1 - r_v^2) = \frac{1}{\alpha_x \alpha_y \theta_{\text{eff}}^4}. \tag{26}
$$

Hence a nontrivial solution exists iff $\alpha_x \alpha_y \theta_{\text{eff}}^4 > 1$, equivalently $\alpha_x \alpha_y \rho^2 \theta^4 > 1$, yielding (10). Solving in the supercritical regime gives

$$
\begin{aligned}
r_u^2 &= \frac{\alpha_x \alpha_y \theta_{\text{eff}}^4 - 1}{\alpha_x \alpha_y \theta_{\text{eff}}^4 + \alpha_y \theta_{\text{eff}}^2} = \frac{\alpha_x \alpha_y \rho^2 \theta^4 - 1}{\alpha_y \rho \theta^2 (\alpha_x \rho \theta^2 + 1)}, \\
r_v^2 &= \frac{\alpha_x \alpha_y \theta_{\text{eff}}^4 - 1}{\alpha_x \alpha_y \theta_{\text{eff}}^4 + \alpha_x \theta_{\text{eff}}^2} = \frac{\alpha_x \alpha_y \rho^2 \theta^4 - 1}{\alpha_x \rho \theta^2 (\alpha_y \rho \theta^2 + 1)},
\end{aligned}
\tag{27}
$$

which finalizes the results of Theorem 3.2. $\qquad\square$

## 4. Experimental Validation

We validate Theorem 3.2 through Monte Carlo simulations across multiple experimental designs, followed by semi-synthetic experiments using real biological data. Code to reproduce all experiments is publicly available. [1]

---

[1] See supplementary material for an anonymized code.

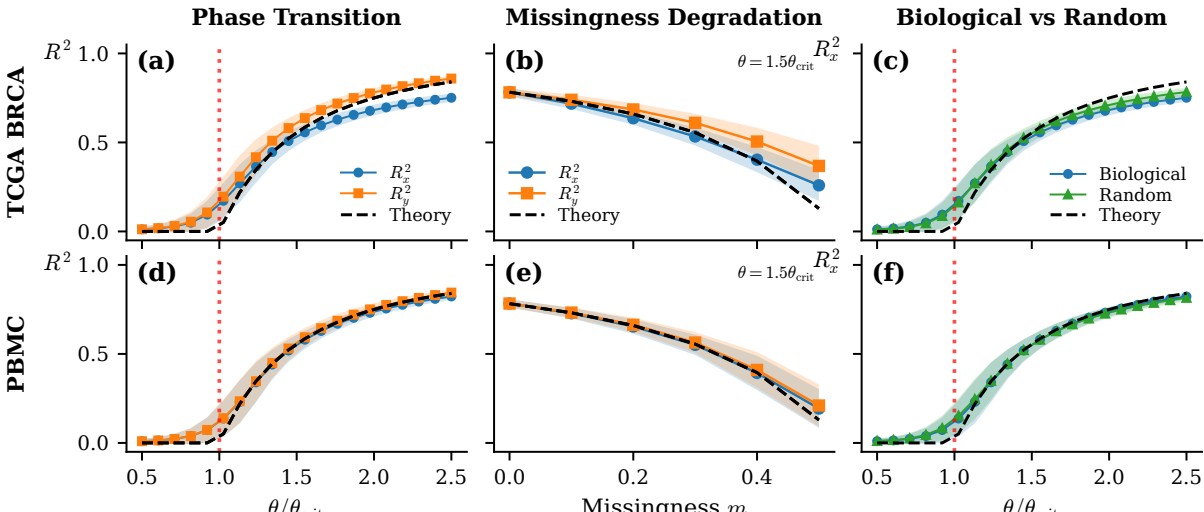

*Figure 3.* **Semi-synthetic validation with biological signal structure.** Top row: TCGA BRCA (cancer genomics, $N = 873$); Bottom row: PBMC Multiome (single-cell, $N = 5000$). **(a, d)** Phase transition curves show empirical overlaps (points with shaded bands showing $\pm 1$ standard deviation) matching theoretical predictions (dashed lines). The transition occurs at $\theta/\theta_{\text{crit}} = 1$ (red line). **(b, e)** Recovery degrades with increasing missingness, following theory (dashed lines). Fixed $\theta = 1.5\theta_{\text{crit}}$. **(c, f)** Biological signal directions (blue) and random Gaussian directions (green) yield identical phase transitions, demonstrating universality. Theory-empirical correlation $r > 0.99$ for both datasets. All experiments use 500 trials per configuration.

## 4.1. Simulation Protocol

For each configuration, we generate a whitened design $X_\star$ via QR decomposition ensuring $X_\star^\top X_\star = N I_{D_x}$, construct the response $Y_\star = \theta(X_\star u_0)v_0^\top + Z$ with i.i.d. Gaussian noise $Z_{ij} \sim \mathcal{N}(0, 1)$, and apply independent MCAR masks $S_x$, $S_y$ with specified retention probabilities. We extract the leading singular vectors $(\hat{u}, \hat{v})$ from the pre-whitened cross-covariance $C = (N\sqrt{\rho})^{-1}X^\top Y$ and compute empirical squared overlaps $R_x^2 = (\hat{u}^\top u_0)^2$ and $R_y^2 = (\hat{v}^\top v_0)^2$.

## 4.2. Synthetic Experiments

We design four synthetic experiments targeting different aspects of the phase transition theory.

**Experiment 1: Phase Transition Validation.** We fix $N = 1000$, $D_x = 200$, $D_y = 50$ (aspect ratios $\alpha_x = 5$, $\alpha_y = 20$), and missingness rates $m_x = 0.3$, $m_y = 0.4$, yielding joint retention $\rho = 0.42$ and critical threshold $\theta_{\text{crit}} \approx 0.49$. We sweep $\theta$ from $0.5\theta_{\text{crit}}$ to $2.5\theta_{\text{crit}}$ across 25 values with 100 independent trials each, measuring both $R_x^2$ and $R_y^2$.

**Experiment 2: Phase Diagram.** To validate the full phase structure, we construct a $30 \times 30$ grid over $\theta \in [0.5, 2.0]$ and $\rho \in [0.1, 0.95]$ with $N = 1000$, $D_x = 150$, $D_y = 120$, running 30 trials per grid point. We compare the empirical $R_x^2$ heatmap against the theoretical phase boundary $\theta_{\text{crit}}(\rho) = 1/[(\alpha_x \alpha_y)^{1/4}\sqrt{\rho}]$.

**Experiment 3: Finite-Size Effects.** We examine convergence to the asymptotic theory by varying sample size $N \in \{100, 250, 500, 1000, 2000, 5000\}$ with fixed aspect ratios $\alpha_x = \alpha_y = 2.5$ and symmetric missingness $m_x = m_y = 0.2$ ($\rho = 0.64$). For each $N$, we sweep $\theta$ in a narrow window around $\theta_{\text{crit}}$ ($\pm 15\%$) with 30 trials per point, observing how the transition sharpens with increasing dimensionality.

**Experiment 4: Missingness Comparison.** We compare single-view versus joint missingness using $N = 800$, $D_x = D_y = 200$ ($\alpha_x = \alpha_y = 4$). For each condition, we construct a $50 \times 50$ grid over $\theta \in [0.3, 2.0]$ and $m \in [0.0, 0.9]$ with 30 trials per point. In the single-view condition, only $X$ is masked ($m_Y = 0$, $\rho = 1 - m$); in the joint condition, both views are masked equally ($m_X = m_Y = m$, $\rho = (1 - m)^2$).

## 4.3. Semi-Synthetic Experiments

The synthetic simulations validate Theorem 3.2 under idealized conditions where the signal directions $(u_0, v_0)$ are independent random Gaussian vectors. However, biological signals often exhibit structured, non-random geometry arising from underlying biological processes. We test whether the phase transition theory holds when signal directions are extracted from real multi-view biological data.

**Semi-Synthetic Protocol.** We construct semi-synthetic data by combining real signal geometry with controlled noise. Given a real multi-view dataset $(X_{\text{real}}, Y_{\text{real}})$:

*Table 1.* Biological datasets for semi-synthetic validation.

| Dataset | $N$ | View $X$ | View $Y$ | Context |
|---------|-----|----------|----------|---------|
| TCGA BRCA | 873 | RNA-seq | Methylation | Cancer |
| PBMC 10k | 5000 | scRNA | scATAC | Single-cell |

1. Preprocess each view: standardize features, apply PCA to reduce to $D_x = D_y = 200$ dimensions, and whiten to satisfy $X_w^\top X_w = N I_{D_x}$.

2. Extract empirical signal directions $(u_{\text{bio}}, v_{\text{bio}})$ as the leading PLS-SVD singular vectors of the whitened data.

3. Generate semi-synthetic responses $Y_\star = \theta(X_w u_{\text{bio}})v_{\text{bio}}^\top + Z$ with controlled Gaussian noise $Z_{ij} \sim \mathcal{N}(0, 1)$ and known signal strength $\theta$.

4. Apply MCAR masks with specified missingness rates $m_x = m_y = m$.

This protocol preserves the biological signal geometry while providing ground-truth directions for computing overlaps.

**Datasets.** We use two multi-view biological datasets representing distinct data modalities and biological contexts (Table 1).

**TCGA BRCA** (The Cancer Genome Atlas, Breast Invasive Carcinoma): RNA sequencing gene expression ($\sim$20,000 genes) paired with DNA methylation profiles ($\sim$450,000 CpG sites) from 873 breast cancer tumor samples (The Cancer Genome Atlas Network, 2012). After preprocessing, the dataset yields $N = 873$ samples with $D_x = D_y = 200$ whitened dimensions.

**PBMC Multiome** (10x Genomics): Single-cell RNA sequencing (scRNA-seq) paired with single-cell chromatin accessibility (scATAC-seq) from peripheral blood mononuclear cells (10x Genomics, 2021). We subsample to $N = 5000$ cells and reduce to $D_x = D_y = 200$ dimensions via PCA on RNA and latent semantic indexing (LSI) on ATAC peaks.

**Experiment 5: Semi-Synthetic Validation.** For each dataset, we conduct three experiments with 500 independent trials per configuration:

*(i) Phase transition validation:* Sweep $\theta$ from $0.5\theta_{\text{crit}}$ to $2.5\theta_{\text{crit}}$ (20 points) at fixed missingness $m = 0.3$, comparing empirical overlaps to theoretical predictions.

*(ii) Missingness degradation:* Fix $\theta = 1.5\theta_{\text{crit}}$ (supercritical regime) and sweep $m \in \{0, 0.1, 0.2, 0.3, 0.4, 0.5\}$, measuring how recovery degrades with increasing missingness.

*(iii) Biological vs. random directions:* Compare phase transitions using biological directions $(u_{\text{bio}}, v_{\text{bio}})$ versus random

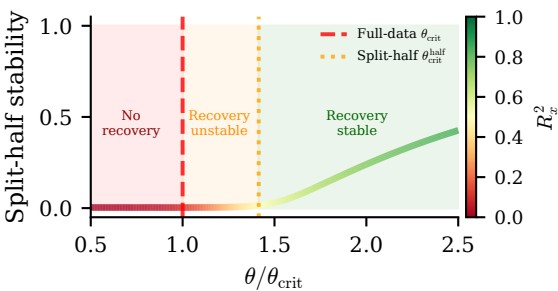

*Figure 4.* **Split-half stability as a practical diagnostic for the phase transition.** Stability (correlation between singular vectors estimated from independent data splits) distinguishes three recovery regimes. **Red dashed line**: full-data critical threshold $\theta_{\text{crit}}$. **Orange dotted line**: split-half threshold $\theta_{\text{crit}}^{\text{half}} = \sqrt{2}\,\theta_{\text{crit}}$ (each half uses $N/2$ samples, halving the aspect ratios). **Shaded regions**: no recovery ($\theta < \theta_{\text{crit}}$, red), recovery unstable ($\theta_{\text{crit}} < \theta < \sqrt{2}\,\theta_{\text{crit}}$, orange), and recovery stable ($\theta > \sqrt{2}\,\theta_{\text{crit}}$, green). Line color indicates true recovery quality $R_x^2$: the gradient from red (subcritical) to green (supercritical) shows that observable stability tracks the theoretically-predicted phase boundaries. This diagnostic is computable without access to ground-truth signal directions. Parameters: $N = 2000$, $\alpha_x = \alpha_y = 7.5$, $m_x = m_y = 0.1$, 25 trials per point.

Gaussian directions $(u_{\text{rand}}, v_{\text{rand}})$ to test universality.

**Experiment 6: Practical Diagnostics.** We investigate whether the phase transition is observable without access to ground-truth signal directions. Using $N = 2000$, $\alpha_x = \alpha_y = 7.5$ (i.e., $D_x = D_y = 266$), and $m_x = m_y = 0.1$, we sweep $\theta$ from $0.5\theta_{\text{crit}}$ to $2.5\theta_{\text{crit}}$ across 60 values with 25 trials each. For each trial, we compute split-half stability: randomly partition samples into two halves, run PLS-SVD on each, and measure the correlation between the resulting singular vectors. This diagnostic is computable in practice without knowing $(u_0, v_0)$.

See Appendix B for experiments on robustness analysis.

## 5. Results

### 5.1. Synthetic Validation

Figure 1 presents the core synthetic validation of Theorem 3.2. Panel (a) demonstrates the phase transition: empirical overlaps $R_x^2$ and $R_y^2$ closely track theoretical predictions, with the sharp transition occurring precisely at $\theta/\theta_{\text{crit}} = 1$ (red vertical line). The asymmetric overlaps $R_x^2 > R_y^2$ reflect the asymmetric aspect ratios ($\alpha_x = 5$ vs. $\alpha_y = 20$), as predicted by the formulas in Eq. (9).

Panel (b) validates the full phase structure across the $(\theta, \rho)$ parameter space. The theoretical phase boundary $\theta_{\text{crit}}(\rho) = 1/[(\alpha_x \alpha_y)^{1/4}\sqrt{\rho}]$ separates the subcritical regime ($R_x^2 \approx 0$) from the supercritical regime ($R_x^2 > 0$). The correlation between predicted and observed overlaps is $r = 0.994$,

confirming quantitative accuracy across the entire parameter space.

Panel (c) examines finite-size effects. As $N$ increases from 100 to 5000, the empirical transition sharpens from a smooth function toward the theoretical step function, confirming that Theorem 3.2 describes the limiting behavior and that finite-sample corrections diminish with increasing dimensionality.

### 5.2. Missingness Effects

Figure 2 compares how single-view versus joint missingness affect the recovery threshold. In panel (a), only view $X$ is masked while $Y$ remains complete. The retention probability degrades linearly as $\rho = 1 - m$, and the phase boundary rises gradually with increasing $m$.

In panel (b), both views are masked equally ($m_X = m_Y = m$). The joint retention probability now degrades quadratically as $\rho = (1-m)^2$, causing a steeper phase boundary. At $m = 0.5$, for instance, single-view masking yields $\rho = 0.5$ while joint masking yields $\rho = 0.25$, requiring twice the signal strength for equivalent recovery.

Joint missingness, common in multi-omics studies where both modalities may suffer from dropout or measurement failures, imposes a significantly higher burden on signal strength than single-view missingness alone.

### 5.3. Semi-Synthetic Validation

Figure 3 presents results for both biological datasets. Panels (a) and (d) show phase transition curves: empirical overlaps $R_x^2$ and $R_y^2$ closely track theoretical predictions, with the transition occurring at $\theta/\theta_{\text{crit}} = 1$. The theory-empirical correlation exceeds $r > 0.99$ for both datasets, despite the signal directions being extracted from real biological covariance structure rather than random Gaussian vectors.

Panels (b) and (e) display missingness degradation: at fixed supercritical signal strength ($\theta = 1.5\theta_{\text{crit}}$), recovery quality decreases smoothly as missingness increases from $m = 0$ to $m = 0.5$, following the theoretical prediction that higher $m$ raises the effective threshold via $\rho = (1 - m)^2$.

Panels (c) and (f) compare biological versus random signal directions. Both curves overlap and follow the same theoretical prediction. This confirms that the phase transition is quite robust to the specific geometry of $(u_0, v_0)$. Only the signal-to-noise ratio $\theta/\theta_{\text{crit}}$ matters.

### 5.4. Practical Diagnostics

Figure 4 demonstrates that the phase transition is observable in practice without access to ground-truth signal directions. Split-half stability, the correlation between singular vectors

estimated from independent random partitions of the data, provides a computable diagnostic that distinguishes three recovery regimes.

A key insight is that split-half analysis uses only $N/2$ samples per half, effectively halving the aspect ratios $\alpha_x$ and $\alpha_y$. This shifts the effective threshold from $\theta_{\text{crit}}$ to $\theta_{\text{crit}}^{\text{half}} = \sqrt{2}\,\theta_{\text{crit}} \approx 1.41\,\theta_{\text{crit}}$. This creates three distinct regimes: (1) *no recovery* ($\theta < \theta_{\text{crit}}$): signal too weak for any recovery; (2) *recovery unstable* ($\theta_{\text{crit}} < \theta < \sqrt{2}\,\theta_{\text{crit}}$): full data recovers signal, but split-halves are unreliable; (3) *recovery stable* ($\theta > \sqrt{2}\,\theta_{\text{crit}}$): both full data and split-halves recover reliably.

The color gradient shows that observable stability closely tracks the true recovery quality $R_x^2$. This finding has practical implications: practitioners can use split-half stability to assess not only whether recovery is possible, but also whether it is robust to data subsampling.

## 6. Conclusion

We studied when spectral Partial Least Squares (PLS-SVD) remains informative for paired multimodal data when *both* views suffer entry-wise MCAR missingness. By showing that dual masking primarily attenuates the effective cross-view spike to $\theta_{\text{eff}} = \sqrt{\rho}\,\theta$, we reduce missing-data PLS to a spiked rectangular random-matrix model with a BBP-style transition. This yields a sharp threshold $\theta_{\text{crit}} = 1/((\alpha_x \alpha_y)^{1/4} \sqrt{\rho})$: below it, the leading singular vectors carry no asymptotic information, while above it they achieve closed-form nonzero overlaps with the planted directions. Extensive simulations confirm the predicted phase diagram and finite-size sharpening, and semi-synthetic experiments on TCGA BRCA and PBMC Multiome show the same transition even with biologically structured signal directions, supporting universality. Finally, split-half stability provides a practical, ground-truth-free diagnostic for operating above the phase boundary. These results show that missingness limits two-view spectral learning through multiplicative signal attenuation and offer simple guidance on the signal strength needed to recover shared structure, motivating extensions to non-MCAR mechanisms, multiple factors, and non-Gaussian noise (see Appendix B for robustness analysis).

## Impact Statement

This paper characterizes when spectral PLS can and cannot recover shared signal from multimodal data under missingness, helping practitioners avoid unstable or spurious components. Misuse in settings where missingness is systematic (not random) could lead to misleading conclusions, so applications should include sensitivity checks and external validation.

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

# A. Technical Details

This appendix provides technical details for the replica derivation: (i) the dual-masking reduction that yields the effective spike $\theta_{\text{eff}} = \sqrt{\rho}\,\theta$, (ii) the Gaussian disorder average identity, (iii) the Gram-matrix Jacobian/entropy factors, (iv) a stationary-point differentiation identity used to extract overlaps from the RS free energy, (v) detailed $w \to 0$ limit calculations, (vi) zero-temperature limit analysis, and (vii) susceptibility optimization.

**Remark on theoretical rigor.** The expressions in Theorem 3.2 are obtained via a replica-symmetric (RS) calculation, which provides a closed-form prediction for both the critical threshold and the asymptotic overlaps. While RS arguments are not universally rigorous, here they match the standard BBP-type behavior expected for spiked random-matrix problems and yield consistent limits in relevant special cases. Crucially, the paper does not rely on these formulas as unchecked claims: Section 4 validates the predictions with extensive simulations over broad parameter sweeps, showing tight quantitative agreement between theory and empirical overlaps. In this sense, Theorem 3.2 should be read as a precise, testable characterization with strong empirical support, and a useful guide for when spectral PLS succeeds or provably fails in practice.

## A.1. Dual MCAR masking and the effective spike strength

We provide additional justification for Lemma 3.1. Recall the latent complete model

$$X_\star^\top X_\star = N I_{D_x}, \qquad Y_\star = \theta(X_\star u_0)v_0^\top + Z, \quad Z_{ij} \sim \mathcal{N}(0,1),$$

and the observed (missing-as-zero) matrices

$$X = S_x \odot X_\star, \qquad Y = S_y \odot Y_\star,$$

with independent MCAR masks having retention probabilities

$$\rho_x = 1 - m_x, \qquad \rho_y = 1 - m_y, \qquad \rho = \rho_x \rho_y.$$

The observed cross-covariance is

$$\widehat{\Sigma}_{XY} = \frac{1}{N} X^\top Y, \qquad C = \frac{1}{\sqrt{\rho}} \widehat{\Sigma}_{XY}.$$

Expanding yields a signal part and a noise part:

$$C = \frac{\theta}{N\sqrt{\rho}} \left(S_x \odot X_\star\right)^\top \left(S_y \odot (X_\star u_0)v_0^\top\right) + \frac{1}{N\sqrt{\rho}} \left(S_x \odot X_\star\right)^\top \left(S_y \odot Z\right).$$

**Mean signal.** Using independence and $\mathbb{E}\left[S_x \odot S_y\right] = \rho$ entrywise,

$$\mathbb{E}\left[C\right] = \frac{\theta}{N\sqrt{\rho}} \mathbb{E}\left[\left(S_x \odot X_\star\right)^\top \left(S_y \odot (X_\star u_0)v_0^\top\right)\right] = \sqrt{\rho}\,\theta\, u_0 v_0^\top.$$

Thus the deterministic spike amplitude is $\theta_{\text{eff}} = \sqrt{\rho}\,\theta$.

**Noise scaling.** Conditioned on $(X, S_y)$, each column of $(S_y \odot Z)$ is Gaussian with covariance $\text{diag}(S_y)$. Multiplication by $X^\top$ yields a (conditionally) Gaussian matrix with covariance controlled by $X^\top X$. Since $X = S_x \odot X_\star$, one has $X^\top X \approx \rho_x N I$ in the proportional limit, and masking in $Y$ contributes an additional factor $\rho_y$, leading to entrywise variance of order $\rho/N$. The normalization by $\sqrt{\rho}$ in $C$ therefore produces an effective noise level $1/\sqrt{N}$, matching the standard spiked rectangular model

$$C = \theta_{\text{eff}}\, u_0 v_0^\top + \frac{1}{\sqrt{N}} W, \qquad W_{ij} \overset{\text{iid}}{\sim} \mathcal{N}(0,1),$$

up to negligible (lower-order) terms.

## A.2. Gaussian Disorder Average

We establish the identity used to average over the Gaussian noise matrix $W$.

**Lemma A.1** (Gaussian disorder average). *For i.i.d. $W_{ij} \sim \mathcal{N}(0,1)$ and order parameters $(Q_u)_{ab} = u^{a\top}u^b$, $(Q_v)_{ab} = v^{a\top}v^b$,*

$$\mathbb{E}_W\left[\exp\left(\beta\sqrt{N}\sum_{a=1}^{w} u^{a\top}Wv^a\right)\right] = \exp\left(\tfrac{\beta^2 N}{2}\,\mathrm{Tr}(Q_u Q_v^\top)\right).$$

*Proof.* Write $\sum_a u^{a\top}Wv^a = \sum_{i,j} W_{ij}A_{ij}$ with $A_{ij} := \sum_a u_i^a v_j^a$. By the Gaussian moment-generating function and independence across entries,

$$\mathbb{E}\left[\exp\left(\beta\sqrt{N}\sum_{i,j} W_{ij}A_{ij}\right)\right] = \prod_{i,j}\mathbb{E}\left[\exp(\beta\sqrt{N}\,W_{ij}A_{ij})\right] = \exp\left(\tfrac{\beta^2 N}{2}\sum_{i,j} A_{ij}^2\right).$$

Computing the sum of squares:

$$\sum_{i,j} A_{ij}^2 = \sum_{i,j}\left(\sum_a u_i^a v_j^a\right)^2 = \sum_{a,b}\left(\sum_i u_i^a u_i^b\right)\left(\sum_j v_j^a v_j^b\right) = \sum_{a,b}(Q_u)_{ab}(Q_v)_{ab} = \mathrm{Tr}(Q_u Q_v^\top).$$

$\square$

### A.3. Jacobian/entropy factors from Gram-matrix variables

We justify the entropic factors $\det(Q_u - r_u r_u^\top)^{D_x/2}$ and $\det(Q_v - r_v r_v^\top)^{D_y/2}$ that arise when changing variables from the replica vectors to the order parameters. Decompose each replica as

$$u^a = (r_u)_a\, u_0 + \tilde{u}^a, \qquad \tilde{u}^a \perp u_0,$$

and stack the orthogonal components into $\tilde{U} = (\tilde{u}^1, \dots, \tilde{u}^w) \in \mathbb{R}^{(D_x-1)\times w}$. Then

$$(Q_u)_{ab} = (u^a)^\top u^b = (r_u)_a (r_u)_b + (\tilde{u}^a)^\top \tilde{u}^b \quad \Rightarrow \quad \tilde{Q}_u := Q_u - r_u r_u^\top = \tilde{U}^\top \tilde{U}.$$

After averaging over the Gaussian disorder, the integrand depends on $\tilde{U}$ only through the Gram matrix $\tilde{Q}_u = \tilde{U}^\top\tilde{U}$ (rotational invariance in the $(D_x-1)$-dimensional subspace orthogonal to $u_0$). Hence we can integrate out the orientation degrees of freedom and use the standard Gram-matrix (Wishart) Jacobian: for $n = D_x - 1$,

$$\int_{\mathbb{R}^{n\times w}} (\cdots)\, d\tilde{U} = \mathrm{const}\cdot\int_{\tilde{Q}_u \succ 0} (\cdots)\,\det(\tilde{Q}_u)^{\frac{n-w-1}{2}}\,d\tilde{Q}_u,$$

where the ellipses denote the same integrand rewritten in terms of $\tilde{Q}_u$ and the multiplicative constant does not depend on $\tilde{Q}_u$. Substituting $\tilde{Q}_u = Q_u - r_u r_u^\top$ gives the factor

$$\det(Q_u - r_u r_u^\top)^{\frac{D_x - 1 - w - 1}{2}} = \exp\left(\frac{D_x}{2}\log\det(Q_u - r_u r_u^\top) + O(1)\right) \propto \det(Q_u - r_u r_u^\top)^{D_x/2},$$

to leading exponential order in $D_x$ (with $w$ fixed). Therefore,

$$\int dU \prod_a \delta(\|u^a\|^2 - 1)\,\delta(Q_u - U^\top U)\,\delta(r_u - U^\top u_0) \propto \det(Q_u - r_u r_u^\top)^{D_x/2},$$

up to multiplicative constants and subexponential factors. The same argument applies to $V$, yielding $\det(Q_v - r_v r_v^\top)^{D_y/2}$.

### A.4. Source Fields and Overlap Extraction

We provide details on the role of source fields and the extraction of overlaps via differentiation.

**Role of the source fields.** The source fields $h_x, h_y$ in the Gibbs partition function (11) serve two purposes:

1. **Symmetry breaking.** The SVD problem (5) has sign ambiguity: if $(\hat{u}, \hat{v})$ is optimal, so is $(-\hat{u}, -\hat{v})$. Without source fields, $\mathbb{E}\left[\hat{u}^\top u_0\right]$ vanishes by symmetry. The limit $h_x, h_y \downarrow 0^+$ selects the positively aligned solution.

2. **Overlap extraction.** The overlaps can be computed by differentiating the free energy with respect to the source fields.

**Overlap extraction via differentiation.** Recall the Gibbs partition function (main text)

$$Z_\beta(h_x, h_y) = \int \delta(\|u\|^2 - 1)\, \delta(\|v\|^2 - 1) \exp\Big(\beta N\, u^\top C v + h_x\, u^\top u_0 + h_y\, v^\top v_0\Big)\, du\, dv.$$

Define $E(u, v) := \beta N\, u^\top C v + h_x\, u^\top u_0 + h_y\, v^\top v_0$. Since $\frac{\partial E}{\partial h_x} = u^\top u_0$, differentiation yields

$$\frac{\partial}{\partial h_x} \log Z_\beta(h_x, h_y) = \big\langle u^\top u_0 \big\rangle_{\beta, h_x, h_y},$$

where $\langle \cdot \rangle_{\beta, h_x, h_y}$ denotes expectation under the Gibbs distribution.

**Value function and stationary-point differentiation.** Define the value function as the RS free energy evaluated at its stationary maximizer:

$$f_\beta(h_x, h_y) := \max_{(q_u, q_v, r_u, r_v) \in \mathcal{D}} \Phi_\beta(q_u, q_v, r_u, r_v; h_x, h_y),$$

where $\mathcal{D}$ is the domain of valid order parameters and $\Phi_\beta$ is given by (21). Let $(q_u^\star, q_v^\star, r_u^\star, r_v^\star) \in \arg\max \Phi_\beta(\cdot; h_x, h_y)$ denote the optimizer. Differentiating $f_\beta$ with respect to $h_x$ by the chain rule gives $\frac{df_\beta}{dh_x} = \sum_j \frac{\partial \Phi_\beta}{\partial p_j} \frac{\partial p_j^\star}{\partial h_x} + \frac{\partial \Phi_\beta}{\partial h_x}$, where $p = (q_u, q_v, r_u, r_v)$. Since $p^\star$ is a stationary point, $\frac{\partial \Phi_\beta}{\partial p_j}\big|_{p^\star} = 0$ for all $j$, so the implicit terms vanish. Moreover, $h_x$ enters $\Phi_\beta$ only through the term $h_x r_u$, so $\frac{\partial \Phi_\beta}{\partial h_x} = r_u^\star$. Therefore,

$$\frac{\partial f_\beta}{\partial h_x} = r_u^\star(h_x, h_y), \qquad \frac{\partial f_\beta}{\partial h_y} = r_v^\star(h_x, h_y).$$

**Limiting overlaps.** The squared overlaps are obtained by taking limits in the following order:

$$R_x^2 = \left( \lim_{h_x, h_y \downarrow 0^+} \lim_{\beta \to \infty} \frac{\partial}{\partial h_x} \mathbb{E}\left[\log Z_\beta(h_x, h_y)\right] \right)^2,$$

$$R_y^2 = \left( \lim_{h_x, h_y \downarrow 0^+} \lim_{\beta \to \infty} \frac{\partial}{\partial h_y} \mathbb{E}\left[\log Z_\beta(h_x, h_y)\right] \right)^2.$$

First, $\beta \to \infty$ concentrates the Gibbs measure on the SVD solution. Then, $h_x, h_y \downarrow 0^+$ removes the symmetry-breaking field while selecting the positive-overlap branch. Consequently, $r_u^\star, r_v^\star$ coincide with the limiting overlaps, yielding

$$R_x^2 = (r_u^\star)^2, \qquad R_y^2 = (r_v^\star)^2.$$

**Explicit verification.** We verify the stationarity-based differentiation identity explicitly. The free energy with source fields has the structure

$$\Phi_\beta(h_x, h_y) = \max_{q_u, q_v, r_u, r_v} \Big\{ \mathcal{E}(q_u, q_v, r_u, r_v) + h_x\, r_u + h_y\, r_v \Big\},$$

where $\mathcal{E}$ collects the entropy, energy, and signal terms (all independent of $h_x, h_y$). Taking the total derivative of $\Phi_\beta$ with respect to $h_x$:

$$\frac{d\Phi_\beta}{dh_x} = \frac{\partial \mathcal{E}}{\partial r_u} \frac{\partial r_u^\star}{\partial h_x} + r_u^\star + h_x \frac{\partial r_u^\star}{\partial h_x}.$$

At the stationary point, $\frac{\partial \mathcal{E}}{\partial r_u} + h_x = 0$. Therefore:

$$\frac{d\Phi_\beta}{dh_x} = \underbrace{\left( \frac{\partial \mathcal{E}}{\partial r_u} + h_x \right)}_{= 0} \frac{\partial r_u^\star}{\partial h_x} + r_u^\star = r_u^\star.$$

By stationarity of the optimizer, the implicit dependence on $h_x$ vanishes, confirming that the saddle-point magnetization equals the typical overlap.

### A.5. Detailed $w \to 0$ Limit Calculations

We provide the step-by-step derivation of the replica limit $w \to 0$ for each term in the action.

**Replica-symmetric structure.** Under the RS ansatz, the $w \times w$ Gram matrix $Q_u$ has the form
$$(Q_u)_{aa} = 1, \quad (Q_u)_{ab} = q_u \text{ for } a \neq b,$$
and the magnetization vector has $(r_u)_a = r_u$ for all $a$. The reduced Gram matrix $\tilde{Q}_u := Q_u - r_u r_u^\top$ has entries:
$$(\tilde{Q}_u)_{aa} = 1 - r_u^2,$$
$$(\tilde{Q}_u)_{ab} = q_u - r_u^2 \quad (a \neq b).$$
This can be written as $\tilde{Q}_u = (1 - q_u)I_w + (q_u - r_u^2)\mathbf{1}\mathbf{1}^\top$, where $\mathbf{1} = (1, \ldots, 1)^\top$.

**Eigenvalue calculation.** A matrix of the form $M = aI + b\mathbf{1}\mathbf{1}^\top$ has eigenvalues:

- $a$ with multiplicity $w - 1$ (eigenvectors orthogonal to $\mathbf{1}$);

- $a + wb$ with multiplicity 1 (eigenvector $\mathbf{1}$).

With $a = 1 - q_u$ and $b = q_u - r_u^2$, the eigenvalues of $\tilde{Q}_u$ are:

- $1 - q_u$ with multiplicity $w - 1$;

- $1 - q_u + w(q_u - r_u^2)$ with multiplicity 1.

The determinant is therefore
$$\det(\tilde{Q}_u) = (1 - q_u)^{w-1}\big(1 - q_u + w(q_u - r_u^2)\big).$$

**Determinant term:** $w \to 0$ **limit.** Taking the logarithm:
$$\log \det(\tilde{Q}_u) = (w - 1)\log(1 - q_u) + \log\big(1 - q_u + w(q_u - r_u^2)\big).$$
Expanding the second term around $w = 0$ using $\log(A + wB) = \log A + \frac{wB}{A} + O(w^2)$:
$$\log\big(1 - q_u + w(q_u - r_u^2)\big) = \log(1 - q_u) + \frac{w(q_u - r_u^2)}{1 - q_u} + O(w^2).$$
Combining:
$$\log \det(\tilde{Q}_u) = w \log(1 - q_u) + \frac{w(q_u - r_u^2)}{1 - q_u} + O(w^2).$$
Therefore:
$$\lim_{w \to 0} \frac{1}{w} \log \det(Q_u - r_u r_u^\top) = \log(1 - q_u) + \frac{q_u - r_u^2}{1 - q_u}. \tag{28}$$
The same formula holds for $v$ by symmetry.

**Trace term:** $w \to 0$ **limit.** Since $\text{Tr}(Q_u Q_v^\top) = \sum_{a,b}(Q_u)_{ab}(Q_v)_{ab}$, under RS:
$$\text{Tr}(Q_u Q_v^\top) = \sum_{a=1}^{w}(Q_u)_{aa}(Q_v)_{aa} + \sum_{a \neq b}(Q_u)_{ab}(Q_v)_{ab}$$
$$= w \cdot 1 \cdot 1 + w(w - 1) \cdot q_u q_v$$
$$= w + w(w - 1)q_u q_v.$$
Therefore:
$$\lim_{w \to 0} \frac{1}{w} \text{Tr}(Q_u Q_v^\top) = 1 + (w - 1)q_u q_v \Big|_{w \to 0} = 1 - q_u q_v. \tag{29}$$

**Signal term:** $w \to 0$ **limit.** Under RS, $(r_u)_a = r_u$ and $(r_v)_a = r_v$ for all $a$, so:
$$\sum_{a=1}^{w}(r_u)_a(r_v)_a = w r_u r_v.$$
Therefore:
$$\lim_{w \to 0} \frac{1}{w} \sum_a (r_u)_a(r_v)_a = r_u r_v. \tag{30}$$

### A.6. Zero-Temperature Limit and Susceptibilities

We derive the zero-temperature ($\beta \to \infty$) limit of the RS free energy. Since $\Phi_\beta$ contains terms like $\frac{\beta^2}{2}(1 - q_u q_v)$ that diverge as $\beta \to \infty$, we extract the ground-state variational problem by computing $\Psi := \lim_{\beta \to \infty} \frac{1}{\beta} \Phi_\beta$. As $\beta \to \infty$, the replicas become identical, forcing $q_u, q_v \to 1$. To avoid indeterminate forms, we introduce the susceptibilities $\chi_u := \beta(1 - q_u)$ and $\chi_v := \beta(1 - q_v)$, which remain $O(1)$ in this limit.

**Entropy contribution.** Consider the entropy term $\frac{1}{\beta} \cdot \frac{1}{2\alpha_x}\left(\log(1 - q_u) + \frac{q_u - r_u^2}{1 - q_u}\right)$. The logarithm $\log(1 - q_u) = \log(\chi_u/\beta)$ grows only as $O(\log \beta)$, so after dividing by $\beta$ it vanishes. For the ratio term, we have

$$\frac{q_u - r_u^2}{1 - q_u} = \frac{(1 - \chi_u/\beta) - r_u^2}{\chi_u/\beta} = \beta \frac{1 - r_u^2}{\chi_u} - 1,$$

which after dividing by $\beta$ contributes $\frac{1 - r_u^2}{\chi_u}$. Thus the entropy term yields $\frac{1 - r_u^2}{2\alpha_x \chi_u}$ in the limit.

**Noise and signal contributions.** For the noise term, expanding $1 - q_u q_v = 1 - (1 - \chi_u/\beta)(1 - \chi_v/\beta) = (\chi_u + \chi_v)/\beta + O(\beta^{-2})$ gives $\frac{\beta^2}{2}(1 - q_u q_v) = \frac{\beta(\chi_u + \chi_v)}{2} + O(1)$, so after dividing by $\beta$ this contributes $\frac{\chi_u + \chi_v}{2}$. The signal term $\beta \theta_{\text{eff}} r_u r_v$ simply becomes $\theta_{\text{eff}} r_u r_v$ after the rescaling.

**Assembled zero-temperature objective.** Combining all contributions:

$$\Psi = \frac{1 - r_u^2}{2\alpha_x \chi_u} + \frac{1 - r_v^2}{2\alpha_y \chi_v} + \frac{\chi_u + \chi_v}{2} + \theta_{\text{eff}} r_u r_v. \tag{31}$$

### A.7. Optimization over Susceptibilities

The structure of (31) allows us to optimize over $\chi_u$ and $\chi_v$ independently.

**Optimizing over $\chi_u$.** Consider the $\chi_u$-dependent terms:

$$f(\chi_u) := \frac{1 - r_u^2}{2\alpha_x \chi_u} + \frac{\chi_u}{2}.$$

Taking the derivative and setting to zero:

$$\frac{df}{d\chi_u} = -\frac{1 - r_u^2}{2\alpha_x \chi_u^2} + \frac{1}{2} = 0 \quad \Rightarrow \quad \chi_u^2 = \frac{1 - r_u^2}{\alpha_x} \quad \Rightarrow \quad \chi_u^\star = \sqrt{\frac{1 - r_u^2}{\alpha_x}}.$$

Substituting back:

$$f(\chi_u^\star) = \frac{1 - r_u^2}{2\alpha_x \chi_u^\star} + \frac{\chi_u^\star}{2} = \frac{\chi_u^\star}{2} + \frac{\chi_u^\star}{2} = \chi_u^\star = \frac{\sqrt{1 - r_u^2}}{\sqrt{\alpha_x}}.$$

**Result for $\chi_v$.** By the same calculation with $(\alpha_y, r_v)$:

$$\chi_v^\star = \sqrt{\frac{1 - r_v^2}{\alpha_y}}, \qquad g(\chi_v^\star) = \frac{\sqrt{1 - r_v^2}}{\sqrt{\alpha_y}}.$$

### A.8. Self-Averaging and Replica Symmetry Validity

**Self-averaging.** In the proportional limit $N, D_x, D_y \to \infty$, the free energy density $\frac{1}{N} \log Z(h_x, h_y)$ concentrates around its expectation. Specifically, fluctuations are of order $O(1/\sqrt{N})$ and thus negligible. This *self-averaging* property justifies replacing the quenched average $\mathbb{E}[\log Z]$ with a saddle-point evaluation of $\log \mathbb{E}[Z^w]$ and replacing sample-dependent quantities with their expectations. Self-averaging is a standard assumption in replica analyses, rigorously justified for many spin glass and random matrix models.

## B. Additional Robustness Experiments

This appendix presents additional experiments examining the robustness of the phase transition theory to violations of key modeling assumptions. We investigate (B.1) robustness to non-Gaussian noise distributions, (B.2) robustness to missing-at-random (MAR) mechanisms that violate MCAR, and (B.3) comparison with alternative missing-data methods.

### B.1. Robustness to Non-Gaussian Noise

Our analysis assumes Gaussian noise in the response matrix $Y_\star$. We test whether the phase transition location and overlap predictions remain accurate under heavy-tailed and heteroskedastic noise distributions.

**Experimental setup.** We fix $N = 1000$, $D_x = 200$, $D_y = 150$ (aspect ratios $\alpha_x = 5$, $\alpha_y = 6.67$), and missingness rates $m_x = m_y = 0.3$ ($\rho = 0.49$), running 100 independent trials per configuration. We compare six noise distributions with unit variance: (1) **Gaussian:** $Z_{ij} \sim \mathcal{N}(0, 1)$ (baseline, excess kurtosis $\kappa = 0$); (2) **Student-$t$ ($\nu = 5$):** standardized ($\kappa = 6$); (3) **Student-$t$ ($\nu = 4.5$):** standardized ($\kappa = 12$); (4) **Student-$t$ ($\nu = 3$):** standardized ($\kappa = \infty$); (5) **Laplace:** standardized ($\kappa = 3$); (6) **Heteroskedastic:** $Z_{ij} \sim \mathcal{N}(0, \sigma_j^2)$ with $\sigma_j^2 \sim \mathrm{Uniform}[0.5, 1.5]$.

**Results.** Figure 5 presents the results. **(a)** Phase transition curves for $R_x^2$ show that all noise types exhibit a sharp transition near the theoretical threshold $\theta_{\mathrm{crit}}$. The transition location is robust, though recovery curves for Student-$t(\nu = 3)$ lie below the Gaussian baseline in the supercritical regime. **(b)** Phase transition curves for $R_y^2$ display analogous behavior. **(c)** Deviation from theory (mean absolute error $|R_x^2 - r_x^2|$ for $\theta > 1.1\theta_{\mathrm{crit}}$) grows with excess kurtosis but remains bounded below 5% but for Student-$t(\nu = 3)$.

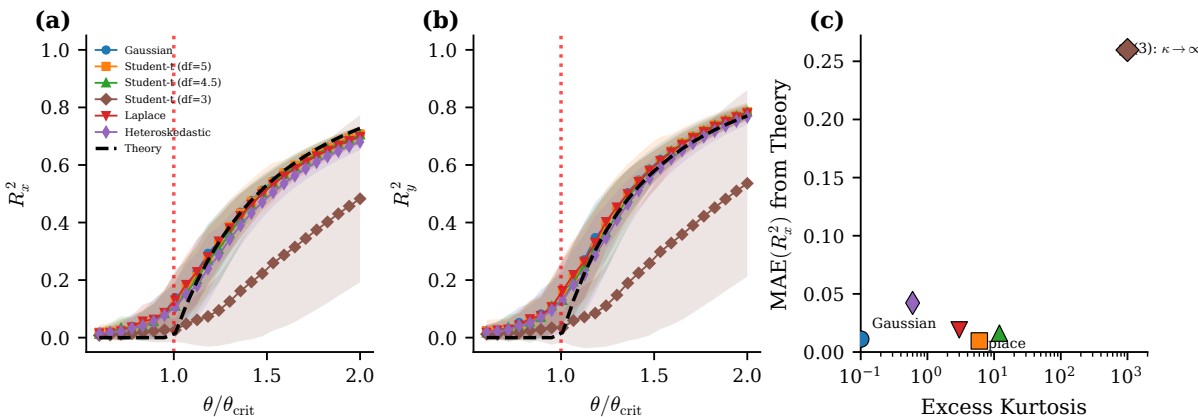

*Figure 5.* **Robustness of phase transition to non-Gaussian noise. (a)** Phase transition in $R_x^2$ across noise types. All distributions exhibit a transition near the critical threshold $\theta_{\mathrm{crit}}$ (red line). Only the Student-$t(\nu = 3)$, which has infinite kurtosis, shows noticeably reduced recovery in the supercritical regime. **(b)** Corresponding phase transition in $R_y^2$. **(c)** Theory deviation versus excess kurtosis. Deviation grows with kurtosis but remains bounded. Parameters: $N = 1000$, $D_x = 200$, $D_y = 150$, $m_x = m_y = 0.3$, 100 trials.

### B.2. Robustness to MAR Mechanisms

Our analysis assumes MCAR missingness, where the mask is independent of the data. We test robustness when missingness depends on the observed values (MAR mechanisms).

**MAR mechanisms.** We consider four mechanisms that create dependence between missingness and data values: (1) **Signal-dependent:** missing probability increases with $|X_\star u_0|$, the projection onto the signal direction; (2) **Magnitude-dependent:** missing probability increases with $|X_{\star,ij}|$; (3) **Thresholded:** entries above a threshold are missing with higher probability; (4) **Correlated:** missingness in $Y$ depends on $X$ values. All mechanisms are calibrated to achieve marginal missing rate $m = 0.3$ on average.

**Experimental setup.** We use $N = 1000$, $D_x = 200$, $D_y = 150$, with MAR strength $\gamma$ varying from 0 (MCAR) to 1 (strong MAR), running 100 trials per configuration.

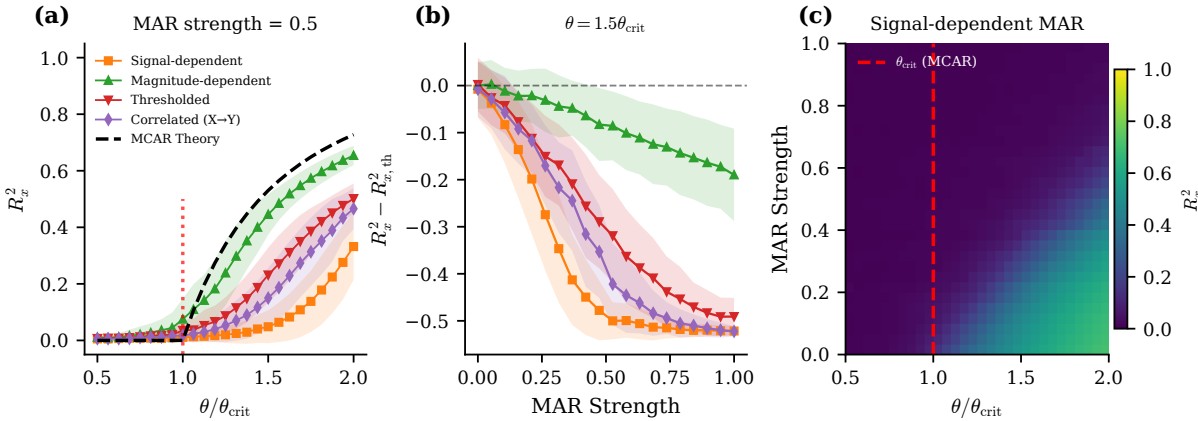

*Figure 6.* **Robustness of phase transition to MAR missingness mechanisms. (a)** Phase transition curves under four MAR mechanisms at strength $\gamma = 0.5$. All exhibit transitions near the MCAR threshold (red line), with signal-dependent MAR showing the most degradation. **(b)** Theory deviation versus MAR strength. Signal-dependent MAR is most harmful; magnitude-dependent MAR has minimal effect. **(c)** Recovery heatmap for signal-dependent MAR across $(\theta/\theta_{\text{crit}}, \gamma)$. The phase boundary shifts with increasing $\gamma$. Parameters: $N = 1000$, $D_x = 200$, $D_y = 150$, target $m = 0.3$, 100 trials.

**Results.** Figure 6 presents the results. **(a)** Phase transition curves under different MAR mechanisms at strength $\gamma = 0.5$. The transition location remains near $\theta_{\text{crit}}$ for all mechanisms, though signal-dependent MAR causes the most degradation in the supercritical regime. **(b)** Deviation from MCAR theory as a function of MAR strength. Signal-dependent MAR is most harmful (deviation $\approx 0.5$ at full strength); magnitude-dependent MAR is most benign (deviation $< 0.2$). **(c)** 2D heatmap of $R_x^2$ across $(\theta, \gamma)$ for signal-dependent MAR. The phase boundary (red dashed) shifts upward with increasing $\gamma$.

The MCAR theory provides a reasonable approximation even under moderate MAR violations. The phase boundary location is robust because it depends primarily on the marginal retention probability $\rho$ rather than the precise missingness mechanism. Signal-dependent MAR is most harmful because it preferentially removes high-signal samples, reducing effective signal strength beyond the $\sqrt{\rho}$ attenuation predicted by MCAR theory.

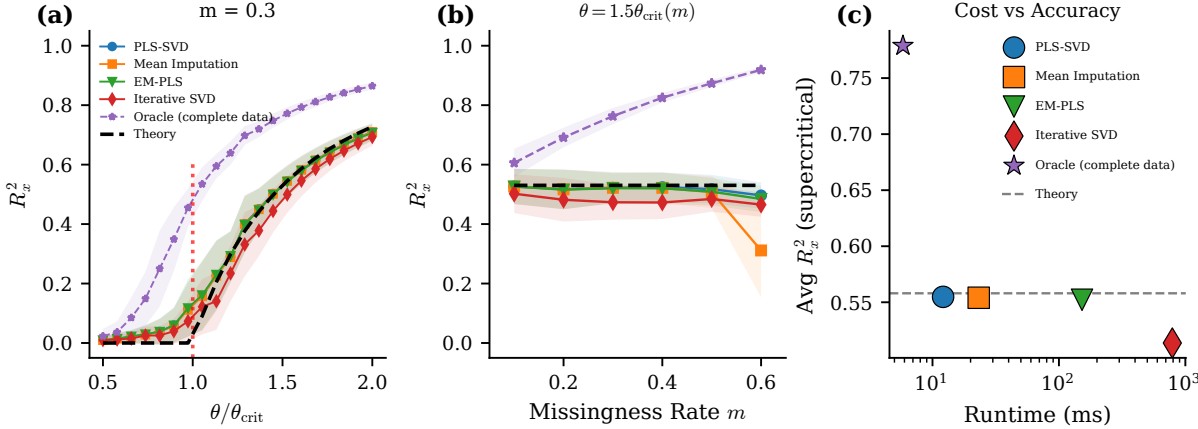

*Figure 7.* **Optimality of simple imputation under MCAR. (a)** Phase transition curves. PLS-SVD (blue) matches theory (dashed); no imputation method exceeds the theoretical limit. Oracle (purple) shows the complete-data upper bound. **(b)** Recovery versus missingness rate at $\theta = 1.5\theta_{\text{crit}}$. All methods track the theoretical prediction. **(c)** Runtime versus accuracy. PLS-SVD achieves the theoretical limit with minimal computation; sophisticated methods add cost without benefit. Parameters: $N = 1000$, $D_x = 200$, $D_y = 150$, $m_x = m_y = 0.3$, 50 trials.

### B.3. Optimality of Simple Imputation

A natural question is whether sophisticated imputation methods can improve upon the simple missing-as-zero approach analyzed theoretically. We test whether more complex methods can exceed the theoretical prediction.

**Methods.** (1) **PLS-SVD:** Missing-as-zero with $C = (N\sqrt{\rho})^{-1}X^\top Y$. (2) **Mean Imputation:** Replace missing entries with column means, then standard PLS-SVD. (3) **EM-PLS:** Iterative EM algorithm alternating between imputation and PLS estimation. (4) **Iterative SVD:** Low-rank SVD imputation followed by PLS-SVD. (5) **Oracle:** PLS-SVD on the complete (unmasked) data $X_\star, Y_\star$ (upper bound).

**Experimental setup.** We use $N = 1000$, $D_x = 200$, $D_y = 150$, with $m_x = m_y = 0.3$, sweeping $\theta$ from $0.5\theta_{\text{crit}}$ to $2.0\theta_{\text{crit}}$ with 50 trials per configuration.

**Results.** Figure 7 presents the results. **(a)** Phase transition curves for all methods. PLS-SVD closely matches the theoretical prediction. Mean Imputation exhibits similar performance. EM-PLS and Iterative SVD do not improve over PLS-SVD—no method exceeds the theoretical limit. The Oracle shows the irreducible cost of missingness. **(b)** Recovery $R_x^2$ as a function of missingness rate $m$ at fixed supercritical signal $\theta = 1.5\theta_{\text{crit}}$. **(c)** Runtime versus accuracy at $\theta = 1.5\theta_{\text{crit}}$, $m = 0.3$. PLS-SVD achieves the theoretical prediction at the lowest computational cost ($\approx$12ms). EM-PLS is $\approx$12$\times$ slower with no accuracy benefit; Iterative SVD is $\approx$65$\times$ slower.

The simple missing-as-zero strategy with proper normalization achieves the theoretical limit for MCAR missingness. Sophisticated imputation methods cannot improve recovery because: (1) MCAR missingness is uninformative about the missing values, and (2) the $\sqrt{\rho}$ normalization correctly accounts for signal attenuation. The gap to the Oracle quantifies the irreducible cost of missingness.

