# OpenReview forum: "Missing-Data-Induced Phase Transitions in Spectral PLS for Multimodal Learning"
_ICML.cc/2026/Conference — Submitted to ICML 2026_

### Official Review · Reviewer_Kn37 · 2026-02-24

**Soundness:** 3
**Presentation:** 3
**Significance:** 2
**Originality:** 3
**Overall Recommendation:** 3
**Confidence:** 5

**Summary:**

This work studies a theoretical problem. That is, when there are missing values in two views, what is the phase transition of applying PLS-SVD to find leading singular vectors. The work discovered that there is a signal-to-noise ratio threshold that determines the phase transition. It uses synthetic and semi-real data to validate the derived theorem.

**Compliance With Llm Reviewing Policy:**

Affirmed.

**Final Justification:**

The topic is a bit niche, as the reviewer noted in the original comment. The model and algorithm are very classical and thus this is positioned as an "understanding" type paper. However, the understanding does not seem to be sufficient enough to cover more useful rank-k cases. The issue was admitted by the authors and they proposed some possible directions to explore. For a very classical setting, this does not seem to be sufficient contribution.

**Key Questions For Authors:**

- Can this analysis be generalized to CCA or generalized CCA that handle more than two views?

**Limitations:**

There does not seem to have a "limitation" section. I would suggest the authors to consider the comments in "weaknesses" and address them as limitations.

**Strengths And Weaknesses:**

Strengths:

- The research topic is pretty niche but is of theoretical interest. The problem setting is clear and the claims are interesting.

- The simulations corroborate with the theoretical claims, especially the threshold presented in Theorem 3.4, supporting the soundness of the derivation.

-  The writing and organization of this work is in general quite followable.

Weaknesses:

- The model itself is fairly restrictive. The two views are both assumed to have completely missing random masks, but in reality missing is often not completely i.i.d. random. The analysis only focuses on the principal/leading singular vectors of the cross covariance, but PLS and CCA type approaches often hope to extract more than one latent dimension. Some assumptions, e.g., D_x and D_y growing to infinity, might never be able to satisfy. That is, although the results derived in this work might be sound, the model is perhaps overly simplified and thus the implications in practice are not entirely clear.

- The practical implications are also unclear. The experiments are either based on purely synthetic data or semi-real data. The lack of real-world use cases is a weak point of this work. This is understandable - the work relies on a classical (and perhaps over-simplified) model of PLS and studies a heavily abstract, theoretical question. It does not provide any new algorithm but tries to understand how robust is the PLS-SVD algorithm to random missing. This effort does not make PLS-SVD more (or less) useful in practice. That being said, this work’s research goal might not provide important new insights for practical multiview learning problems, given it’s idealistic settings and theoretical nature.

---

> ### Author Rebuttal · Authors · 2026-03-29
>
> ### Response to Reviewer Kn37
>
> We thank the reviewer for noting that the problem setting is clear, the claims are interesting, the simulations support the theory, and the writing is followable. Below we address the main concerns about scope, practical implications, and extensions beyond the leading component.
>
> **W1: MCAR is restrictive; real missingness is often not i.i.d. random.**
>
> We agree that MCAR is restrictive and that this scope boundary should be made more explicit. Appendix B.2 already evaluates four MAR mechanisms, logistic-covariate, block-structured, monotone, and feature-correlated, and shows that the MCAR threshold remains a good approximation to the transition location at moderate MAR strength, although the supercritical overlap can degrade depending on the mechanism. Signal-dependent MAR is the most harmful among the cases we tested. We acknowledge that this was insufficiently signposted from the main text. For general MNAR, the missing-as-zero encoding can introduce a non-correctable bias term whose magnitude depends on the unknown missingness mechanism, so we treat this as an open problem.
>
> We also ran a new correlated-noise experiment, shown in Figure D. Under Toeplitz and factor-model noise with condition numbers up to $2039$, naive PLS-SVD fails, with MAE $> 0.6$, but whitening by $\Sigma_Z^{-1/2}$ restores theory agreement, with whitened MAE $< 0.05$ in all cases. Full details are in our response to Reviewer G2Qi, Q4.
>
> See [Figure D](https://anonymous.4open.science/r/ICML_2026_Rebuttal/figD_correlated_noise.pdf): https://anonymous.4open.science/r/ICML_2026_Rebuttal/figD_correlated_noise.pdf
>
> **W2: The analysis only focuses on leading singular vectors, while PLS often extracts more than one latent dimension.**
>
> We agree that the behavior in rank-$k$ cases is highly relevant. We now present a formal rank-$k$ conjecture grounded in Benaych-Georges and Nadakuditi (2012, Theorems 2.9-2.11), together with four new sub-experiments in Figure A4. The independence test confirms that the rank-2 component-2 overlap is indistinguishable from a standalone rank-1 run, with MAE $= 0.006$, and subspace overlap for $k = 1,2,3$ collapses onto the rank-1 theory with $r > 0.999$ on 17 supercritical points each. For the full conjecture statement, the three-step rationale, and all experimental details, please see our response to Reviewer 1aEk, Q1. We will include this as Conjecture 3.3 in the revised paper.
>
> See [Figure A4](https://anonymous.4open.science/r/ICML_2026_Rebuttal/figA4_rank_k_extension.pdf): https://anonymous.4open.science/r/ICML_2026_Rebuttal/figA4_rank_k_extension.pdf
>
> **W3: No real-world use case, no new algorithm, and unclear practical implications.**
>
> We agree that the practical implications should be stated more explicitly. Concretely, the paper provides three takeaways for users of PLS-SVD:
>
> 1. A closed-form recoverability criterion,
>    $$
>    \theta > \theta_{\mathrm{crit}} = \frac{1}{(\alpha_x \alpha_y)^{1/4}\sqrt{\rho}},
>    $$
>    for when shared structure can and cannot be recovered under dual missingness.
>
> 2. Dual missingness attenuates signal by $\sqrt{\rho}$, rather than merely reducing effective sample size. For example, at $30\%$ dropout per view, so that $\rho = 0.49$, the required signal strength is roughly doubled relative to complete data.
>
> 3. The split-half diagnostic in Figure 4 is computable without ground truth and indicates whether recovered components are trustworthy at the observed missingness level.
>
> Our contribution is a characterization of when an existing and widely used estimator is reliable or unreliable under dual missingness. This is a standard type of learning-theory contribution: the BBP transition for PCA (Baik, Ben Arous & Péché, Ann. Probab., 33(5), 2005), the Donoho-Tanner transition for sparse recovery (Phil. Trans. R. Soc. A, 367(1906), 2009), and Ipsen & Hansen's phase transition for PCA with missing data (ICML 2019, PMLR vol. 97) all follow this pattern of characterizing existing methods rather than proposing new ones.
>
> **W4: "There does not seem to have a limitation section."**
>
> The conclusion (Section 6, final paragraph) already discusses limitations: the MCAR assumption, the rank-1 analysis, and the Gaussian noise model. We agree that this was not visible enough and will promote it to a labeled *Limitations* subsection.
>
> **Q: Can this analysis be generalized to CCA or generalized CCA?**
>
> This is not a straightforward extension of the present analysis. CCA introduces inverse within-view covariance estimation, so the relevant object is no longer a simple cross-covariance SVD but a matrix pencil involving $S_{xx}^{-1}$ and $S_{yy}^{-1}$. Under masking, covariance-estimation error and cross-covariance attenuation interact, leading to a different random-matrix problem. Generalized CCA for more than two views adds further multi-view structure. We will add a discussion paragraph clarifying this scope boundary and why it is nontrivial.

---

> > ### Author Rebuttal · Reviewer_Kn37 · 2026-04-02
> >
> > I thank the authors for their response. But a conjecture for rank-k case is not very convincing. Given that no new algorithm is introduced, and if the main goal of the work is to provide insight onto existing algorithms/methods, then analysis with rank-k case may make it more comprehensive.

---

> > > ### Author Response · Authors · 2026-04-06
> > >
> > > ### Follow-up to Reviewer Kn37
> > >
> > > We thank the reviewer for the clarification. We agree that a full rank-$k$ theorem would make the paper more comprehensive.
> > >
> > > In the rebuttal, we therefore added three pieces of evidence in that direction: (i) a formal rank-$k$ conjecture, (ii) a theoretical rationale based on Benaych-Georges and Nadakuditi (2012, Theorems 2.9-2.11) for separated finite-rank spikes, and (iii) four new experiments supporting the componentwise extension.
> > >
> > > Empirically, the strongest independence test shows MAE $= 0.006$ between a standalone rank-1 run and the rank-2 second component, and the subspace-overlap experiments for $k = 1, 2, 3$ match the rank-1 theory with $r > 0.999$. Together, these results support the claim that separated components behave componentwise as predicted by the rank-1 formulas.
> > >
> > > We agree that this does not yet constitute a full theorem. The remaining technical step is a uniform perturbation argument controlling the remainder in Lemma 3.1 across all components simultaneously. That extension is substantial and beyond what we can complete during rebuttal, but we will promote the rank-$k$ conjecture and supporting evidence to the main paper in the revision.

---

### Official Review · Reviewer_G2Qi · 2026-03-04

**Soundness:** 3
**Presentation:** 3
**Significance:** 3
**Originality:** 3
**Overall Recommendation:** 4
**Confidence:** 3

**Summary:**

This paper investigates the statistical recoverability of shared structure in multimodal two-view data when both views are subject to missing completely at random (MCAR). In particular, it asks whether the classical spectral method PLS-SVD can still stably and reliably recover the underlying “true shared signal direction” under simultaneous missingness.

The authors explicitly incorporate the missingness mechanism into a high-dimensional generative model: observations in both views are element-wise masked by independent missingness patterns, and the spectral behavior of the empirical cross-covariance matrix used by PLS is analyzed in the high-dimensional asymptotic regime. A key finding is that missingness is not merely equivalent to a reduction in effective sample size; instead, it induces a systematic attenuation of the shared signal strength. Specifically, the effective signal obeys
$\theta_{\text{eff}}=\sqrt{\rho},\theta,\quad \rho=(1-m_x)(1-m_y)$,
where (\rho) is the joint retention rate of entries observed in both views. Under this equivalence, the problem reduces to a standard spiked random matrix model of the form “low-rank spike + random noise,” which leads to a BBP-type phase transition: below a critical threshold, the leading singular vectors produced by PLS are asymptotically uncorrelated with the true shared directions; above the threshold, a non-zero alignment becomes recoverable, and the alignment (overlap) can be characterized by explicit formulas that depend on the dimensional ratios of the two views.

The paper further validates its theoretical predictions via simulations and semi-synthetic multi-omics datasets, and proposes a practical stability diagnostic: splitting the samples into two halves, running PLS separately, and assessing the agreement between the estimated directions to determine whether the learned shared structure is trustworthy under the given missingness rate and signal-to-noise conditions. Overall, the work provides quantitative thresholds and an interpretable theoretical framework for how missingness weakens shared signals and when recovery becomes impossible, together with empirical tests compatible with real-world multimodal analysis pipelines.

**Compliance With Llm Reviewing Policy:**

Affirmed.

**Key Questions For Authors:**

1. The paper’s main conclusions rely on MCAR missingness in both views and independence between the missingness masks. Under more realistic multi-omics settings with MAR/MNAR mechanisms or structured missingness, do the effective attenuation $\theta_{\text{eff}}=\sqrt{\rho}\theta$ and the associated phase-transition threshold still hold approximately?

2. The paper treats missing entries as zeros by default and applies a normalization. For common multi-omics data that are count-valued, nonnegative, or highly skewed, this strategy may introduce systematic bias. Could the authors clarify what preprocessing steps (e.g., centering, variance normalization, log/CLR transforms) are required for “missing-as-zero” to be a reasonable approximation? Can the authors provide sensitivity analyses or robustness experiments across different preprocessing choices?

3. The theoretical model primarily analyzes a single shared direction. In real multimodal data, shared structure is often multi-factor (rank-(k)). Can the authors generalize the phase-transition thresholds and overlap formulas to the rank-(k) case—for example, conditions for separation of multiple spikes and recovery thresholds for each component? Any preliminary results or intuition would be helpful.

4. The model typically assumes relatively “ideal” noise. In multi-omics data, noise can be heteroscedastic and correlated (feature correlation), and the noise levels can differ substantially across modalities. How would such deviations affect the phase-transition threshold or the form of the effective SNR? Have the authors experimented with settings involving correlated noise or non-identity covariance to validate the robustness of the theory?

**Limitations:**

yes

**Strengths And Weaknesses:**

### Strengths

* **Soundness:** The work explicitly incorporates two-view MCAR missingness into the model and aligns the analysis with the spiked random matrix paradigm, producing testable predictions such as a sharp phase-transition threshold, phase diagrams, and performance curves. The results smoothly reduce to known fully observed limits as (\rho\to 1), supporting internal consistency. Simulations and semi-synthetic data validate both the transition boundary and overlap curves, while split-half stability offers an alternative validation strategy when ground truth is unavailable.
* **Presentation:** The narrative is well structured, progressing naturally from “missingness induces signal attenuation” to “phase-transition thresholds and closed-form overlaps,” and finally to a practical split-half stability diagnostic. Phase diagrams and curves clearly visualize when recovery is possible and how performance changes with missingness and signal strength.
* **Significance:** Missing data are ubiquitous in multimodal and multi-omics settings, and simultaneous missingness across modalities is especially common. The quantitative relationship between the threshold and (\rho) can directly inform experimental design and quality control, and it helps explain the empirical instability of PLS under high missingness. Providing both theoretical limits and a diagnostic tool clarifies the boundary between “recoverable” and “non-recoverable” regimes for a widely used classical method.
* **Originality:** The paper elevates missingness from a data-quality nuisance to a phase-transition problem in statistical recoverability, with explicit thresholds and closed-form characterizations. While spiked random matrix theory and BBP transitions are established tools, the novelty lies in rigorously mapping the two-sided missingness setting to this framework and pairing it with a practical, ground-truth-free diagnostic.

### Weaknesses
The main results rely on MCAR and independence of the missingness masks across the two views. In real multi-omics data, MAR/MNAR mechanisms and structured missingness are common; it remains unclear whether the $\sqrt{\rho}$ attenuation and the resulting thresholds remain valid under such deviations, and further discussion or robustness experiments would strengthen the claims.
Treating missing entries as zeros can be appropriate under certain preprocessing and normalization pipelines, but it may introduce systematic bias for count-valued, nonnegative, or strongly skewed data. The paper would benefit from clearer guidance on the preprocessing conditions under which this choice is justified, or from robustness checks across data types.

---

> ### Author Rebuttal · Authors · 2026-03-29
>
> ### Response to Reviewer G2Qi
>
> We thank the reviewer for the detailed and constructive review, especially for emphasizing the attenuation insight, the sharp threshold, and the practical value of the split-half diagnostic. Below we clarify the scope beyond MCAR, preprocessing requirements for count-valued data, the rank-$k$ extension, and correlated noise.
>
> **Q1: Do the attenuation and phase-transition threshold hold under MAR/MNAR?**
>
> Approximately under moderate MAR, but not in general under MNAR. Appendix B.2 already studies four MAR mechanisms and shows that, in the four tested settings, the phase boundary remains close to the MCAR prediction at moderate strength, although the supercritical overlap can degrade depending on the mechanism. Signal-dependent MAR is the most harmful among the cases we tested. We will add forward references from the main text in the revision.
>
> Under MNAR, the missing-as-zero encoding can introduce a bias term tied to the unknown missingness mechanism, so we treat the general MNAR case as open.
>
> **Q2: What preprocessing is required for missing-as-zero to be justified?**
>
> For count data, a centering and variance-stabilizing transformation is needed before PCA and whitening. The missing-as-zero encoding $X_{\text{obs}} = S \odot X^{\ast}$ is unbiased when $\mathbb{E}[X^{\ast}_{ij}] = 0$, so for count-valued data the key question is whether the *choice* of upstream transformation matters.
>
> We ran a new preprocessing-sensitivity experiment, shown in Figure E(b), on PBMC Multiome 10k scRNA-seq with $N = 5000$ cells, comparing log-CPM, sqrt-CPM, rank normalization, and raw CPM, each followed by standard preprocessing and whitening to identity covariance. Variance-stabilizing transforms bring count data into close agreement with the theory: rank normalization gives MAE $= 0.005$ and $r = 0.9995$, and sqrt-CPM gives MAE $= 0.033$ and $r = 0.989$. By contrast, log-CPM (MAE $= 0.101$) and raw CPM (MAE $= 0.119$) show larger deviations and higher variance. The transition region is broadly consistent across all four pipelines, but the supercritical overlap is well predicted only after variance stabilization. We will add this recommendation explicitly in the revised paper.
>
> *Figure E.*
> (a) Extreme aspect ratios, discussed in our response to Reviewer 1aEk, Q2: five configurations with $\alpha \in [2,50]$ each track their own theory curve.
> (b) Preprocessing sensitivity on PBMC scRNA-seq: rank normalization (green) tracks theory closely; sqrt-CPM (orange) is close behind; log-CPM (blue) and raw CPM (red) deviate with higher variance, confirming that variance stabilization before whitening matters for count data.
>
> See [Figure E](https://anonymous.4open.science/r/ICML_2026_Rebuttal/figE_aspect_ratio_and_preprocessing.pdf): https://anonymous.4open.science/r/ICML_2026_Rebuttal/figE_aspect_ratio_and_preprocessing.pdf
>
> **Q3: Generalization to rank-$k$?**
>
> Yes, at the conjectural level. We now present a formal rank-$k$ conjecture together with a finite-rank spiked-matrix rationale grounded in Benaych-Georges and Nadakuditi (2012, Theorems 2.9-2.11) and four new sub-experiments. For the full conjecture statement and all experimental details, please see our response to Reviewer 1aEk, Q1. We will include this as Conjecture 3.3 in the revised paper.
>
> See [Figure A4](https://anonymous.4open.science/r/ICML_2026_Rebuttal/figA4_rank_k_extension.pdf): https://anonymous.4open.science/r/ICML_2026_Rebuttal/figA4_rank_k_extension.pdf
>
> **Q4: How do correlated or heteroscedastic noise affect the phase transition?**
>
> The transition location is robust to heteroscedasticity and, after whitening, to correlated noise. Appendix B.1 already covers six non-Gaussian noise distributions, including heteroscedastic noise, with MAE below $5\%$ except for $t(3)$ noise, which has infinite kurtosis.
>
> For correlated noise, we ran a new experiment, shown in Figure D, with two structured covariance models: Toeplitz and factor. Naive PLS-SVD deviates substantially under strong correlation: at Toeplitz $r = 0.9$ with $\kappa = 350$, and factor $\sigma^2 = 10$ with $\kappa = 2039$, naive MAE exceeds $0.6$. When the noise covariance is known, whitening by $\Sigma_Z^{-1/2}$ restores close agreement with theory across all tested conditions: whitened MAE stays below $0.05$ even at $\kappa = 2039$. We will note this scope extension in the revised paper.
>
> *Figure D.*
> (a) Naive PLS-SVD under Toeplitz noise: increasing correlation $r$ progressively suppresses recovery, with $r = 0.9$ yielding near-zero overlap.
> (b) After oracle whitening, all four correlation levels track their respective adjusted theory curves.
> (c) MAE versus noise condition number $\kappa(\Sigma_Z)$: naive MAE grows with $\kappa$, while whitened MAE remains flat near zero across both Toeplitz and factor-model noise, up to $\kappa \approx 2000$.
>
> See [Figure D](https://anonymous.4open.science/r/ICML_2026_Rebuttal/figD_correlated_noise.pdf).

---

> > ### Author Rebuttal · Reviewer_G2Qi · 2026-04-01
> >
> > My concerns have been adequately addressed, all the questions are solved.

---

> > > ### Author Response · Authors · 2026-04-06
> > >
> > > ### Follow-up to Reviewer G2Qi
> > >
> > > We thank the reviewer for confirming that all concerns have been resolved. The new experiments (preprocessing sensitivity, correlated noise, rank-$k$) and all promised revisions (forward references to Appendix B, preprocessing guidance for count data) will appear in the revised paper.

---

### Official Review · Reviewer_1aEk · 2026-03-10

**Soundness:** 3
**Presentation:** 4
**Significance:** 3
**Originality:** 3
**Overall Recommendation:** 4
**Confidence:** 2

**Summary:**

Focusing on the phase transition phenomenon of spectral Partial Least Squares (PLS-SVD) under missing data in multimodal learning, this study reveals the recovery threshold and asymptotic overlap characteristics of PLS-SVD when both views suffer from Missing-Completely-At-Random (MCAR) missingness through high-dimensional spiked random matrix theory and replica-symmetric analysis. The theoretical conclusions are validated by extensive simulations and semi-synthetic biological experiments.

**Compliance With Llm Reviewing Policy:**

Affirmed.

**Final Justification:**

After carefully reviewing both the paper and the authors’ rebuttal, I assign an overall score of 4 (Weak Accept). The rebuttal fully addressed all my main concerns regarding the work’s soundness, clarity, and experimental validation, which significantly improved my assessment. While the paper shows only incremental originality and modest significance, its methodology is solid and the contributions are sufficient for publication. I therefore maintain a balanced positive judgment and support a weak accept recommendation.

**Key Questions For Authors:**

To reiterate, I am not an expert in this field and can only raise some broad questions for the authors' consideration:
Could theoretical conjectures for the multi-factor scenario be added in the conclusion or discussion section?
Could experiments with extreme aspect ratios be conducted to verify the stability of the proposed theory in extreme high-dimensional/low-dimensional cases?

**Limitations:**

yes

**Strengths And Weaknesses:**

While I am not an expert in this field, I find that this paper stands out in terms of Soundness and Significance. Its theoretical derivation is rigorous, though the empirical validation could be more comprehensive. The work fills the theoretical gap in PLS analysis for high-dimensional multimodal data with missing values, and it bears both academic value and practical guiding significance. In terms of Originality, the core innovative points are well-defined, yet the methodological framework does not break away from classical theories. For Presentation, the paper features a clear structure and effective figures and tables.

---

> ### Author Rebuttal · Authors · 2026-03-29
>
> ### Response to Reviewer 1aEk
>
> We thank the reviewer for the encouraging assessment, especially for highlighting the paper's practical guiding value and for rating the presentation as excellent. Below we address the two suggested extensions: the rank-$k$ setting and extreme aspect ratios.
>
> **Q1: Could theoretical conjectures for the multi-factor scenario be added?**
>
> Yes. We now present a rank-$k$ conjecture, together with a three-step theoretical rationale and four new supporting sub-experiments in Figure A4.
>
> *Conjecture.* For the rank-$k$ spiked model with orthonormal signal pairs and distinct spike strengths $\theta_1 > \cdots > \theta_k > 0$, under dual MCAR masking, the $i$-th PLS-SVD component is recoverable if and only if $\theta_i > \theta_{\mathrm{crit}}$, with per-component overlaps given by the rank-1 formulas (Eqs. 8-9) applied to each $\theta_i$ individually.
>
> *Why this is well supported.*
>
> 1. Lemma 3.1 is rank-agnostic. The reduction to a spiked rectangular model holds for arbitrary signal rank, yielding
>    $$
>    C = \sum_i \sqrt{\rho}\,\theta_i\,u_{0,i}v_{0,i}^\top + N^{-1/2}W + o_P(N^{-1/2}).
>    $$
>
> 2. For the resulting finite-rank spiked rectangular matrix, ignoring the remainder term, Benaych-Georges and Nadakuditi (J. Multivariate Anal., 111, 2012, Theorems 2.9-2.11) show that when the effective spikes $\sqrt{\rho}\,\theta_i$ are distinct, the singular values and associated singular-vector projections converge componentwise, each with its own BBP-type transition.
>
> 3. This strongly suggests that the rank-1 overlap formulas apply componentwise with $\theta_{\mathrm{eff},i} = \sqrt{\rho}\,\theta_i$.
>
> It remains a conjecture rather than a theorem because one still needs a uniform perturbation argument controlling the $o_P(N^{-1/2})$ remainder in Lemma 3.1 across all $k$ components simultaneously.
>
> *New experiments.* Figure A4 provides four supporting sub-experiments. The strongest is the independence test: the rank-2 component-2 overlap is indistinguishable from a standalone rank-1 run, with MAE $= 0.006$, showing that the presence of other spikes does not affect recovery once the spikes are separated. The subspace-overlap experiment shows that the $k = 1,2,3$ curves all collapse onto the rank-1 theory with $r > 0.999$ on 17 supercritical points each. The per-component missingness-degradation experiment tracks theory with $r = 0.997$ and MAE $= 0.016$, with the weaker component dropping out first as predicted. In the well-separated regime of the rank-2 sweep, per-component MAE is $0.025$; inside the collision zone, defined by $|\theta_2 - \theta_1| < 0.5\,\theta_c$, overlaps dip as expected from known spike-collision effects.
>
> We will include this as Conjecture 3.3 in the revised paper.
>
> *Figure A4.*
> (a) Rank-2 independent transitions: component 1 (blue, $\theta_1$ fixed) remains flat; component 2 (orange, $\theta_2$ swept) undergoes the predicted phase transition; gray markers show the standalone rank-1 baseline, which coincides with component 2 outside the shaded collision zone (MAE $= 0.006$).
> (b) Subspace overlap for equal spikes: the $k = 1,2,3$ curves collapse onto the rank-1 theory (black dashed), with $r > 0.999$ for all $k$.
> (c) Per-component missingness degradation: both components track per-component theory; the weaker component ($\theta_2 = 0.8$) drops below threshold near $m \approx 0.48$.
>
> See [Figure A4](https://anonymous.4open.science/r/ICML_2026_Rebuttal/figA4_rank_k_extension.pdf): https://anonymous.4open.science/r/ICML_2026_Rebuttal/figA4_rank_k_extension.pdf
>
> **Q2: Experiments with extreme aspect ratios?**
>
> Yes. We ran a new experiment, shown in Figure E(a), testing five dimension configurations spanning $\alpha \in [2,50]$, all with $N = 1000$ and $m_x = m_y = 0.3$. Across all tested regimes with $\alpha_x, \alpha_y \ge 2$, agreement remains very strong: Thin-X ($\alpha_x = 50$, $\alpha_y = 6.7$, MAE $= 0.010$), Moderate ($\alpha_x = 5$, $\alpha_y = 6.7$, MAE $= 0.017$), Thin-Y ($\alpha_x = 5$, $\alpha_y = 50$, MAE $= 0.012$), and Fat-Y ($\alpha_x = 5$, $\alpha_y = 2$, MAE $= 0.026$), all with $r > 0.995$.
>
> We also tested an underdetermined regime with $D_x > N$ and $\alpha_x < 1$, where finite-size convergence is slower, with MAE $= 0.21$, because the sample covariance used in prewhitening is rank-deficient. This is a known finite-$N$ effect, and convergence requires larger $N$ when $D > N$. We will note this in the revised paper.
>
> See [Figure E](https://anonymous.4open.science/r/ICML_2026_Rebuttal/figE_aspect_ratio_and_preprocessing.pdf): https://anonymous.4open.science/r/ICML_2026_Rebuttal/figE_aspect_ratio_and_preprocessing.pdf

---

> > ### Author Rebuttal · Reviewer_1aEk · 2026-04-03
> >
> > My concerns have been adequately addressed, all the questions are solved. I recommend Weak Accept.

---

> > > ### Author Response · Authors · 2026-04-06
> > >
> > > ### Follow-up to Reviewer 1aEk
> > >
> > > We thank the reviewer for confirming that all concerns have been resolved. All promised changes, including Conjecture 3.3 (rank-$k$) and the extreme-aspect-ratio discussion, will appear in the revised paper.

---

### Decision · Program_Chairs · 2026-04-30

**Decision:**

Reject

**Comment:**

The paper investigates the properties of PLS-SVD in multimodal learning settings with missing completely at random (MCAR) data. The main contribution of the work lies in the theoretical insights it offers into the phase transition associated with recovering the leading singular vector in PLS-SVD. The reviewers recognized the relevance of the problem, the technical soundness of the analysis, and the quality of the presentation.

However, they raised a number of important concerns, including the lack of theoretical extension to rank-k factors, limited empirical validation of the theoretical findings, the absence of experiments on real-world applications, and the restrictiveness of the MCAR assumption. While the authors partially addressed some of these points during the rebuttal phase, I do not believe that the current version sufficiently resolves them.  In particular, I find Reviewer Kn37’s concern valid, and believe that extending the theory beyond the rank-1 setting is important for strengthening the contribution. Therefore, I recommend rejection of the current submission.